# RefTool: Reference-Guided Tool Creation for Knowledge-Intensive Reasoning

**Xiao Liu**[1,2]  **Da Yin**[3]  **Zirui Wu**[1]  **Yansong Feng**[1]*
[1] Wangxuan Institute of Computer Technology, Peking University  [2] University of Chicago
[3] University of California, Los Angeles
{lxlisa, ziruiwu, fengyansong}@pku.edu.cn, da.yin9712@gmail.com

## Abstract

Large Language Models (LLMs) can enhance their reasoning capabilities by using external tools. However, many tasks lack predefined tools. Prior works have explored instructing LLMs to generate tools on their own, but such approaches depend heavily on internal knowledge and struggle when tasks fall outside the model's knowledge scope. To address this limitation, we propose REFTOOL, a reference-guided framework for automatic tool creation that leverages external materials, such as textbooks and knowledge snippets. REFTOOL consists of two modules: (1) tool creation, where LLMs generate executable tools from reference content, validate them using illustrative examples, and organize them hierarchically into a toolbox; and (2) tool utilization, where LLMs navigate the toolbox structure to select and apply the appropriate tools to solve problems. Experiments on causality, physics, and chemistry benchmarks demonstrate that REFTOOL outperforms existing tool-creation and domain-specific reasoning methods by 12.3% on average accuracy, while being cost-efficient and broadly generalizable to non-scientific tasks, e.g., extremely low-resource language translation. Analyses reveal that grounding tool creation in references produces accurate and faithful tools, and that the hierarchical structure facilitates effective tool selection. REFTOOL enables LLMs to overcome internal knowledge limitations, advancing generalizable reasoning in knowledge-intensive domains. Code and data are available at https://github.com/xxxiaol/RefTool.

## 1 Introduction

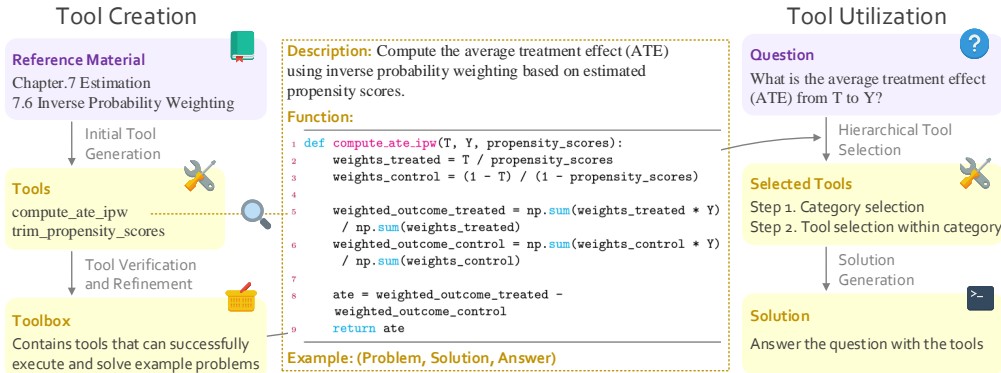

Figure 1: Overview of the REFTOOL framework, which consists of two modules: tool creation (left) and tool utilization (right).

Tools play a critical role in enhancing the reasoning capabilities of large language models (LLMs), particularly in scientific problem-solving like mathematical reasoning (Lu et al., 2023; Zhang et al.,

---

*Corresponding author.

2023). By integrating external tools, LLMs can use off-the-shelf modules to complete subtasks and execute precise computations, thereby improving their performance.

Despite their importance, such tools are not universally available across all scenarios. A prominent line of work attempts to mitigate this limitation by instructing LLMs to generate their own tools based on given problems (Qian et al., 2023; Cai et al., 2023; Wang et al., 2024b). However, these methods would fall short when models lack the relevant expert knowledge, especially in specialized and novel domains. For example, if an LLM is unfamiliar with *how to estimate the causal effect from a treatment variable to an outcome variable*, it can hardly generate appropriate tools for such tasks.

To address this challenge, we propose REFTOOL, a reference-guided framework for automatic tool creation. Unlike existing methods that rely on LLMs' internal knowledge, REFTOOL leverages external reference materials, such as textbooks and knowledge snippets, that naturally cover a broad range of domains. As shown in Figure 1, REFTOOL consists of two modules: tool creation and tool utilization. During tool creation, the framework employs LLMs to generate executable tools from reference content. In the example, given a segment on *Inverse probability weighting*[1] from a causal inference textbook, the LLM produces tools like `compute_ate_ipw` according to the content. The generated tools consist of descriptions, functions, alongside illustrative examples teaching models when and how to use the tools. These examples also serve as validation cases, filtering out non-functional or incorrect tools while retaining those that successfully solve the example problems. The validated tools are organized into a hierarchical toolbox, mirroring the structure of the reference material, or created by the model if the reference is unstructured.

During inference, REFTOOL guides the LLM to select tools from the toolbox hierarchically and apply tools to solve problems. For an input question like *what is the average treatment effect from T to Y*, the LLM navigates the toolbox hierarchy, selecting the *Estimation* category and then the `compute_ate_ipw` tool within the category. Finally, the LLM generates the solution with the help of the selected tool. By grounding tool creation and selection in external references rather than internal knowledge, REFTOOL can construct and deploy tools beyond the model's original capabilities, enabling it to tackle tasks that would otherwise be infeasible.

We evaluate REFTOOL across three knowledge-intensive scientific domains: causality, physics, and chemistry. With the help of textbooks, REFTOOL outperforms existing tool creation methods by 13.0% on average, highlighting the value of incorporating external knowledge in tool creation. REFTOOL also achieves an average accuracy improvement of 10.2% over domain-specific reasoning methods (Pang et al., 2025; Ouyang et al., 2024; Tang et al., 2025). Unlike prior works that depend on manually constructed toolsets or extensive trial-and-error on validation data, REFTOOL achieves greater efficiency in both time and computational cost.

REFTOOL exhibits strong generalization. Akin to human learning knowledge, the generated tools are not dataset-specific, but maintain robust performance across diverse datasets in the domain. REFTOOL further proves effective in non-scientific tasks and with unstructured references. In extremely low-resource language translation, it organizes unstructured grammar rules into a hierarchy and transforms them into pseudo-code tools, yielding improved translation performance.

To summarize, we propose REFTOOL, a reference-guided framework for tool creation. REFTOOL has the following advantages: (1) By leveraging reference materials, REFTOOL enables LLMs to generate tools beyond their internal knowledge. (2) Experiments on diverse knowledge-intensive reasoning tasks demonstrate that REFTOOL consistently improves performance over existing baselines. (3) REFTOOL generates dataset-agnostic tools in a cost-efficient and human-free manner, demonstrating the potential for extending the knowledge boundary of LLMs in real-time problem solving.

## 2 THE REFTOOL FRAMEWORK

REFTOOL operates in two stages: (1) constructing a hierarchical toolbox $T$ from reference material $R$, and (2) selecting and applying tools $t \subset T$ to answer the input question $q$ during inference. This section introduces the method with a focus on scientific reasoning tasks, and §3.3 describes how the method is adapted to non-scientific tasks.

---

[1]Inverse probability weighting is a common method for estimating causal effects.

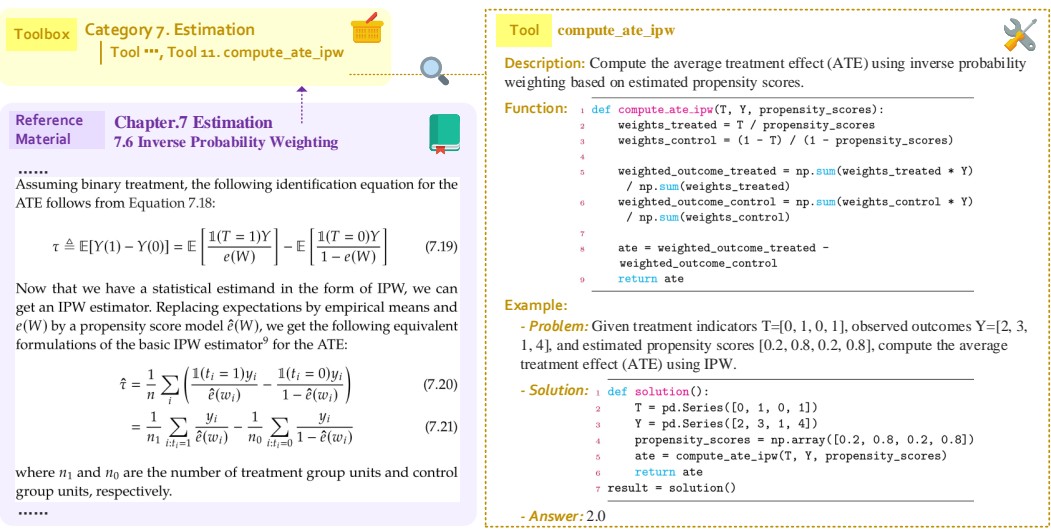

Figure 2: Example of a generated tool and its corresponding reference segment.

## 2.1 THE TOOL CREATION MODULE

**Knowledge Organization** The first step in tool creation is to organize the knowledge from reference materials into a structured form. In this work, we adopt a two-level hierarchy for knowledge organization, which also defines how the generated tools are arranged.

Many references, such as textbooks and technical documents, naturally follow a hierarchical organization that supports systematic knowledge acquisition. At the highest level, they are divided into *chapters* (e.g., `Estimation` in the causal inference textbook), which are further decomposed into *sections* (e.g., `Inverse Probability Weighting`), each addressing a specific technique, theorem, or application within the broader chapter context. For such cases, we directly extract this inherent structure and adopt the chapters as the first-level categories.

For unstructured references like knowledge snippets, we ask an LLM to construct the hierarchy based on the content. The model first proposes category names and then assigns reference segments to appropriate categories. After the conversion from unstructured reference material to structured ones, we apply the same general method introduced in the following to create tools.

**Initial Tool Generation** Given a reference segment, such as a section $s_i \in \mathcal{R}$, the LLM is instructed to generate executable tools based on its content. Each tool consists of three key components, as illustrated in Figure 2. First, a **description** provides a natural language summary of the tool's purpose. Second, a **function** offers a Python implementation of the tool, including comments that explain its parameters and return values. Finally, an **example** demonstrates the tool's usage, comprising a problem, a piece of solution code where the tool is invoked, and the expected answer. The model prioritizes examples from the reference text when available, otherwise generating an appropriate example by itself. The LLM is asked to generate at most $m$ tools for each section. To ensure proper formatting, the prompt includes a human-written tool example from a different domain.

**Tool Verification and Refinement** Each tool is verified through *execution testing* and *output verification* using the model-generated demonstration example. The solution code should run without errors, and the output should match the expected answer. Failed tools trigger a refinement step, where the failure information is provided to the LLM to refine the tool. Finally, the valid tools are organized hierarchically into a toolbox.

## 2.2 THE TOOL UTILIZATION MODULE

**Hierarchical Tool Selection** During inference, REFTOOL performs hierarchical retrieval to select tools for question $q$ through two phases:

Table 1: Statistics of the reference materials and created tools. "Avg. Lines" indicates the average lines of tool functions.

| Domain | Book | # Categories | # Sections | # Tools | Avg. Lines |
|---|---|---|---|---|---|
| Causality | Introduction to Causal Inference (Neal, 2020) | 11 | 55 | 84 | 24 |
| Physics | University Physics (Ling et al., 2016) | 44 | 284 | 515 | 16 |
| Chemistry | Atkins' Physical Chemistry (Atkins et al., 2023) | 19 | 90 | 158 | 17 |

- **Category Selection**: Given the toolbox categories $C$, the model is instructed to select at most $n_c$ relevant categories $c \subset C$ for the question $q$.
- **Tool Selection within Category**: For each selected category $c_i$, the model is given access to all tools from the toolbox $T$ associated with that category, including their descriptions, functions, and demonstration examples. It is then prompted to select up to $n_t$ relevant tools $t$, or none if no tools are deemed applicable.

**Solution Generation** The selected tools are then integrated into the reasoning process. We incorporate the tools with two reasoning paradigms: single-turn Program-of-Thoughts (PoT) reasoning (Chen et al., 2023a) and multi-turn ReAct-style agent reasoning (Yao et al., 2022). For both paradigms, the model receives selected tools in the initial prompt and is instructed to invoke them when appropriate. When no suitable tools are identified, REFTOOL defaults to standard PoT or ReAct reasoning, ensuring graceful degradation for questions outside the reference domain.

## 3 EXPERIMENTS

We conduct experiments on three knowledge-intensive scientific domains: causality, physics, and chemistry. This section introduces the experimental setup, presents the performance of REFTOOL, and validates its generalizability to other datasets and non-scientific domains.

### 3.1 EXPERIMENTAL SETUP

**Datasets** We employ the following evaluation benchmarks: (1) **Causality**: QRData-causal (Liu et al., 2024a), where each question is accompanied by one or multiple datasheets. Models are asked to analyze the datasheets and answer causal questions. (2) **Physics**: TheoremQA-physics (Chen et al., 2023b), covering broad topics of university-level physics. (3) **Chemistry**: SciBench-chemistry (Wang et al., 2024a), focusing on three sub-datasets (chemmc, quan, and matter) related to physical and quantum chemistry.[2]

We maintain consistent evaluation protocols (like answer extraction methods and tolerance rates) with the original benchmarks (see Appendix A.1 for details) and report accuracy as our primary metric.

**Reference Materials** Analogous to humans preparing for an exam by reading relevant textbooks, we select reference materials that have a similar domain of knowledge to the evaluation datasets.

For causality, we choose *Introduction to Causal Inference* (Neal, 2020), which provides a detailed description of main causal inference topics like causal discovery and estimation. For physics, as university physics is a broad domain, we choose the three-volume textbook *University Physics* (Ling et al., 2016), which covers the core concepts of physics like mechanics, thermodynamics, and modern physics. For chemistry, given that the benchmark is in physical chemistry, we choose a famous physical chemistry textbook *Atkins' Physical Chemistry* (Atkins et al., 2023).[3] Table 1 (left) provides detailed statistics. Note that none of the evaluation questions originate from these books, and none of these books contain code directly.

**Implementation Details** We employ GPT-4o (Hurst et al., 2024) for tool creation, and evaluate four prevalent LLMs for tool utilization: Llama-3.1-70B (Dubey et al., 2024), Gemini-1.5-Pro (Team et al., 2024), GPT-4 (OpenAI, 2023), and GPT-4o.

---

[2]We omit the other sub-dataset atkins because its question source overlaps with our reference material.

[3]Quantum chemistry is a subdomain of physical chemistry, and is also introduced in this textbook.

Table 2: Performance comparison in causality (QRData), physics (TheoremQA), and chemistry (SciBench). Numbers are in percentages (%), with the best performance for each model in bold.

| Method | Accuracy | | | | |
|---|---|---|---|---|---|
| | Llama-3.1-70B | Gemini-1.5-Pro | GPT-4 | GPT-4o | Average |
| **Causality** | | | | | |
| LATM | 33.5 | 32.3 | 27.0 | 20.9 | 28.4 |
| Creator | 14.9 | 29.7 | 39.4 | 39.8 | 31.0 |
| TroVE | 23.8 | 33.5 | 34.2 | 36.4 | 32.0 |
| PoT | 33.1 | 41.3 | 34.2 | 39.8 | 37.1 |
| PoT + RAG | 29.7 | 36.4 | 37.5 | 42.0 | 36.4 |
| PoT + REFTOOL | **36.8** | **43.9** | **38.7** | **46.8** | **41.6** |
| ReAct | 30.1 | 47.6 | 50.9 | 46.5 | 43.8 |
| ReAct + RAG | 32.3 | 46.8 | 48.0 | 49.1 | 44.1 |
| ReAct + REFTOOL | **33.5** | **48.3** | **51.3** | **52.0** | **46.3** |
| **Physics** | | | | | |
| LATM | 38.9 | 33.3 | 39.8 | 30.6 | 35.7 |
| Creator | 40.4 | 57.0 | 35.1 | 40.4 | 43.2 |
| TroVE | 33.3 | **58.8** | 35.1 | 48.2 | 43.9 |
| Physics Reasoner | 48.2 | 50.9 | 42.1 | 33.3 | 43.6 |
| PoT | 48.2 | 57.9 | 45.6 | 57.0 | 52.2 |
| PoT + RAG | 44.7 | 57.0 | 44.7 | **57.9** | 51.1 |
| PoT + REFTOOL | **53.5** | **58.8** | **49.1** | **57.9** | **54.8** |
| **Chemistry** | | | | | |
| LATM | 31.4 | 25.1 | 45.0 | 35.7 | 34.3 |
| Creator | 40.1 | 60.0 | 46.9 | 43.3 | 47.6 |
| TroVE | 38.6 | 65.6 | 39.2 | 52.7 | 49.0 |
| StructChem | 37.9 | 50.2 | 29.7 | 40.5 | 39.6 |
| ChemAgent | 48.2 | 65.5 | 52.5 | 58.9 | 56.3 |
| PoT | 46.9 | 62.3 | 51.8 | 58.9 | 55.0 |
| PoT + RAG | 48.1 | 63.7 | **54.1** | 56.6 | 55.6 |
| PoT + REFTOOL | **49.5** | **66.4** | 53.4 | **61.3** | **57.7** |

During tool creation, we set $m = 2$ tools per section across all domains. This can be adjusted based on each section's length and information density. For tool utilization, we employ a default configuration of selecting $n_c = 1$ category and $n_t = 1$ tool. As QRData and TheoremQA do not have a validation set, we use the default setting for the causality and physics domains. For chemistry, we perform grid search over $n_c \in [1, 2]$ and $n_t \in [1, 2]$ on the validation set of SciBench, and choose $n_c = 1$ and $n_t = 2$. Additional details and prompt templates can be found in Appendices A.2 and F.

**Baseline Methods** We compare against the following baselines: (1) *General reasoning methods* including Program-of-Thoughts (PoT) for single-turn reasoning and ReAct for multi-turn reasoning;[4] (2) *Retrieval-augmented generation (RAG) methods* using the same reference books employed for tool creation; (3) *General-purpose tool creation methods* including LATM (Cai et al., 2023), Creator (Qian et al., 2023), and TroVE (Wang et al., 2024b); (4) *Domain-specific reasoning methods* including Physics Reasoner (Pang et al., 2025), StructChem (Ouyang et al., 2024), and ChemAgent (Tang et al., 2025). Detailed descriptions of the baselines are in Appendix A.3.

## 3.2 RESULTS

**Toolbox Construction** Table 1 (right) demonstrates statistics of tools created. On average, 73% of initially generated tools pass validation directly, with an additional 14% tools succeeding after refinement. We assess tool quality through human evaluation in §4.3.

---

[4]While ReAct demonstrates effectiveness on QRData by allowing error correction through multi-turn interactions (Liu et al., 2024a), our preliminary experiments (Appendix Table 7) show limited benefits for physics and chemistry domains, likely due to the simpler code solutions without data analysis and fewer execution errors. Consequently, we omit ReAct for these domains.

**Main Results**   The performance comparison in Table 2 demonstrates REFTOOL's superior performance across all domains, achieving the highest average accuracy.[5] For each of the three domains, REFTOOL's accuracy (aggregated across all four LLMs) is significantly higher than all baseline methods at the significance level $\alpha = 0.05$. Notably, REFTOOL surpasses all tool creation methods by an average margin of $13.0\%$, highlighting the advantage of references in tool creation.

While RAG incorporates the same reference materials, it fails to consistently enhance the performance. This suggests that direct retrieval struggles to effectively extract and apply relevant knowledge, while REFTOOL's tool format and hierarchical organization enable better utilization of reference materials.

Among domain-specific methods, Physics Reasoner and StructChem perform inferior to PoT, with their complex format requirements leading to suboptimal adaptation on some models. Although ChemAgent approaches REFTOOL's performance, it needs significantly higher computational costs, as discussed in §4.2.

**Performance on Reasoning Models**   We also conduct a small-scale experiment to evaluate if REFTOOL works for reasoning models. We apply REFTOOL on o1-mini (Jaech et al., 2024), and Appendix Table 10 shows that its average accuracy improves by $4.3\%$ over PoT. This indicates that REFTOOL is also compatible with reasoning models, supplementing their knowledge and skills.

**Robustness of Tool Creation**   We validate whether REFTOOL remains effective when using alternative LLMs for tool creation. Appendix Table 11 shows that creating tools with Gemini-1.5-Pro and Llama-3.1-70B-Instruct also achieves superior performance compared to baseline methods.

## 3.3   GENERALIZABILITY OF REFTOOL

Table 3: Performance of REFTOOL on extremely low-resource language translation (%).

| BLEU / chrF++ | Zhuang → Chinese | | Chinese → Zhuang | | Average |
| --- | --- | --- | --- | --- | --- |
| | Llama-3.1-70B | Qwen-2.5-72B | Llama-3.1-70B | Qwen-2.5-72B | |
| Whole Grammar Book | 32.6 / 35.2 | 43.7 / 42.6 | 28.0 / 60.8 | 34.9 / 64.2 | 34.8 / 50.7 |
| Retrieval from Whole Book | 33.6 / 35.3 | 45.3 / 43.1 | 26.0 / 54.2 | 39.2 / 66.5 | 36.0 / 49.8 |
| Rule-by-Rule Retrieval | 42.6 / 43.0 | 47.4 / 46.8 | 36.1 / 66.4 | 40.6 / 69.9 | 41.7 / 56.5 |
| Rule-by-Rule Retrieval w. Code | 36.4 / 45.1 | 56.9 / **55.7** | 35.3 / 64.8 | 47.0 / **74.3** | 43.9 / 60.0 |
| REFTOOL | **52.2 / 49.6** | **57.2** / 54.1 | **54.2 / 71.6** | **52.5** / 71.1 | **54.0 / 61.6** |

We validate the generalizability of REFTOOL by examining (1) whether tools created for one dataset can be reused across other datasets in the same domain, and (2) whether the framework remains effective in non-scientific domains and when handling unstructured reference materials.

**Tool Reusability**   We conduct experiments on another physics dataset SciBench-fund (Wang et al., 2024a) to validate the generalizability of tools created. Appendix Table 12 shows that on SciBench-fund, REFTOOL outperforms all zero-shot baseline methods and matches 4-shot Physics Reasoner, using the same tools as in the evaluation of TheoremQA. Since REFTOOL is dataset-agnostic, tools developed for one domain can be readily applied to other datasets within that domain.

**Applying REFTOOL to Extremely Low-Resource Language Translation**   Extremely low-resource (XLR) language translation is a representative non-scientific knowledge-intensive task, where LLMs, lacking prior knowledge of the XLR language, are tasked with translation using resources such as dictionaries, parallel sentences, and grammar rules. Grammar rules are crucial for guiding the translation (Tanzer et al., 2024; Team et al., 2024), and in this experiment, we explore whether REFTOOL can help LLMs better select and apply these rules.

We experiment on Zhuang–Chinese translation using the ZHUANGRULES dataset (Zhang et al., 2025)[6], which provides 109 grammar rules. The rules are presented in an **unstructured** format, making it difficult to identify the relevant ones. To address this, we ask the LLM to organize the rules into a two-level hierarchy. Despite limited prior knowledge of Zhuang, the model leverages its general linguistic knowledge to propose categories such as `Numerals and Quantifiers` and `Word Order and Sentence Structure`.

---

[5]Chemistry results are averaged over three sub-datasets, with sub-dataset performance in Appendix Table 9.

[6]Zhuang is a language spoken by the Zhuang people of Southern China.

Table 4: Ablation results (%). (sim) indicates selecting with text similarity.

| Method | Accuracy | | | | |
| --- | --- | --- | --- | --- | --- |
| | Llama-3.1-70B | Gemini-1.5-Pro | GPT-4 | GPT-4o | Average |
| **Causality** | | | | | |
| PoT + RAG | 29.7 | 36.4 | 37.5 | 42.0 | 36.4 |
| PoT + Hierarchical RAG | **36.8** | 38.7 | 43.5 | 36.8 | 39.0 |
| PoT + REFTOOL (sim) | 30.5 | 36.1 | **46.1** | 39.4 | 38.0 |
| PoT + REFTOOL | **36.8** | **43.9** | 38.7 | **46.8** | **41.6** |
| **Physics** | | | | | |
| PoT + RAG | 44.7 | 57.0 | 44.7 | **57.9** | 51.1 |
| PoT + Hierarchical RAG | 44.7 | **64.0** | 44.7 | 55.3 | 52.2 |
| PoT + REFTOOL (sim) | 45.6 | 62.3 | 43.0 | 56.1 | 51.8 |
| PoT + REFTOOL | **53.5** | 58.8 | **49.1** | **57.9** | **54.8** |

We then apply REFTOOL to create tools for each grammar rule. Since the goal is to facilitate rule understanding rather than execution, the tools are represented as pseudo Python code. During verification, the LLM assesses whether the tool functions correctly apply to examples, rather than real code execution.

We compare with previous XLR translation methods, including prompting LLMs with the *Whole Grammar Book*, *Retrieval from Whole Book*, *Rule-by-Rule Retrieval* (which examines each rule individually), and *Rule-by-Rule Retrieval w. Code* (which converts rules into pseudo code). Consistent with the settings of Zhang et al. (2025), we use GPT-4o to construct and organize the tools, and evaluate performance on two open-source LLMs: Llama-3.1-70B and Qwen-2.5-72B (Yang et al., 2024). More implementation details are in Appendix A.4.

Results in Table 3 showcase that REFTOOL outperforms all baselines on average, with an increase of 10.1% in BLEU and 1.6% in chrF++. By organizing rules hierarchically, creating tools to facilitate understanding of the rules, and verifying and revising the tools for better quality, REFTOOL enhances the translation performance. This highlights the broad applicability of REFTOOL to non-scientific domains and its effectiveness in handling unstructured reference materials.

## 4 ANALYSIS

In this section, we further analyze the effectiveness of REFTOOL through: ablation study of key components (§4.1), cost analysis (§4.2), human evaluation of tool quality (§4.3), and case study of how REFTOOL helps LLMs to answer questions (§4.4).

### 4.1 ABLATION STUDY

We design two variants of REFTOOL to analyze its key components: code-form tool creation and hierarchical selection. (1) *PoT + Hierarchical RAG*: Substitutes REFTOOL's code-form tools with raw text segments while preserving the hierarchical structure. This maintains the three-step reasoning process of category selection, intra-category text retrieval, and solution generation. (2) *PoT + REFTOOL (sim)*: Retains the tool creation but replaces hierarchical selection with similarity-based retrieval. Tool descriptions are encoded into embeddings, with the most similar tool selected for each problem, mirroring standard RAG approaches but using tools instead of text.

Table 4 shows the ablation results. Due to computational constraints, we focus on causality and physics domains with single-turn reasoning. By comparing PoT + REFTOOL with PoT + Hierarchical RAG, as well as PoT + REFTOOL (sim) with PoT + RAG, we observe an average 1.9% accuracy gain of tools over textual knowledge, confirming that the code form of tools enhances model understanding and application of knowledge.

By comparing PoT + REFTOOL with PoT + REFTOOL (sim), as well as PoT + Hierarchical RAG with PoT + RAG, we find that hierarchical selection outperforms similarity-based retrieval by 2.6% on average, demonstrating its effectiveness in knowledge retrieval.

## 4.2 COST ANALYSIS

Table 5: Cost analysis of representative tool-augmented methods (with GPT-4o as the base model). "Human" indicates that the step is done by humans and the cost is unknown.

| Domain | Method | Time (min.) | | Cost ($) | |
|---|---|---|---|---|---|
| | | Toolbox Construction | Inference | Toolbox Construction | Inference |
| Physics | Physics Reasoner | Human | 75 | Human | 3.5 |
| | PoT + REFTOOL | 5 | **2** | 6.9 | **1.5** |
| Chemistry | ChemAgent | 1233 | 536 | 79.3 | 41.3 |
| | PoT + REFTOOL | **3** | **6** | **3.5** | **1.4** |

Table 5 shows that REFTOOL greatly saves cost compared with two representative tool-augmented methods, and the comparison with a broader range of models is in Appendix B.6. Compared with Physics Reasoner which iteratively refines the reasoning process, REFTOOL reduces inference time by 97% and cost by 57%. The improvements are even more pronounced when compared to ChemAgent's divide-and-retry strategy: REFTOOL cuts both toolbox construction time and inference time by 99%. The guidance of references enables REFTOOL to achieve great performance without repeatedly trying, offering a scalable solution for complex reasoning tasks.

## 4.3 HUMAN EVALUATION OF TOOL QUALITY

Table 6: Tool quality assessment (%). Example correctness is evaluated only if the function is correct.

| Domain | Faithful | Function Correct | Example Correct | Useful |
|---|---|---|---|---|
| Causality | 96 | 94 | 96 | 92 |
| Physics | 90 | 94 | 100 | 94 |
| Chemistry | 96 | 92 | 96 | 90 |

We conduct human evaluation to assess the quality of created tools along four dimensions. (1) *Faithfulness*: Whether the tool accurately reflects the source material. (2) *Function Correctness*: Whether the tool function meets the tool description and is implemented correctly. (3) *Example Correctness*: Whether the example solution uses the function properly and returns the right answer. (4) *Usefulness*: The practical utility of the tool for solving relevant problems without being too narrow. We randomly sample 50 tools for each domain from the toolbox, and ask a human expert who has studied corresponding courses to annotate them.

As shown in Table 6, all aspects are satisfied by $\geq 89\%$ tools, indicating that most tools are faithfully derived from the references, correctly implemented, and useful in application. Chemistry tools show slightly lower quality due to the domain's complexity. When LLMs lack foundational knowledge, they may misinterpret nuanced concepts, like mistaking the meaning of a coefficient. We further analyze the alignment between LLM-selected tools and those chosen by domain experts in Appendix C.

## 4.4 CASE STUDY

Figure 3 demonstrates a case where GPT-4o correctly answers a question with REFTOOL. When presented with the causal discovery problem, the model successfully navigates to the relevant category *Causal discovery from observational data* and selects the appropriate `causal_direction_fit` tool, generating the correct solution code. In contrast, without tool assistance, the model incorrectly uses R-squared values to infer causal relationships, leading to a wrong prediction. The detailed version of the case, along with physics and chemistry cases,is in Appendix D.

## 5 RELATED WORK

**Automatic Tool Creation** The automatic creation of tools for LLMs aims to overcome the limitations of relying solely on pre-existing tools. Most works generate tools in code format, while some

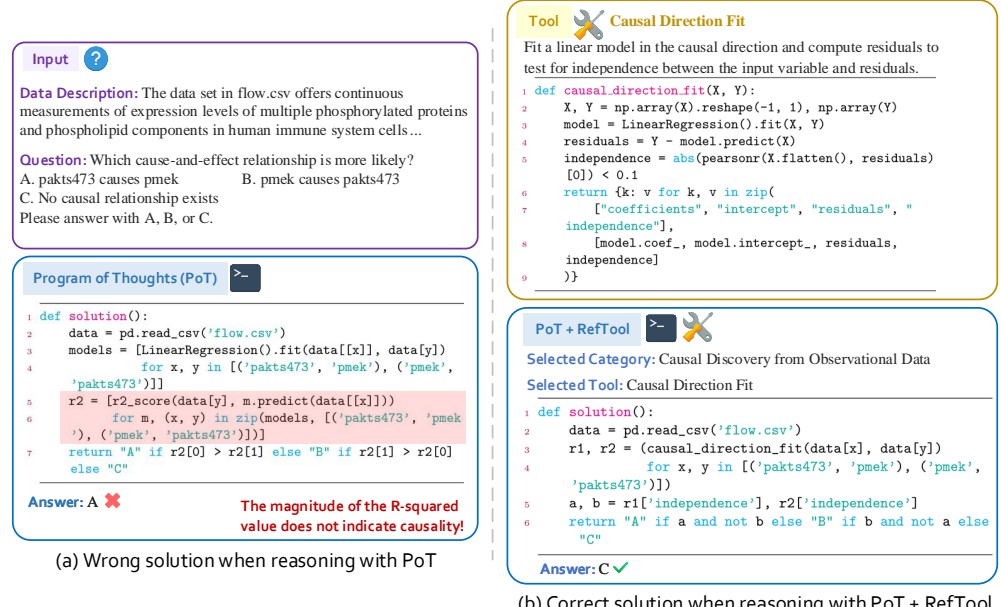

Figure 3: Example case of GPT-4o with (right) and without (left) REFTOOL.

generate skills or workflows in the format of abstract actions (Wong et al., 2024) or non-executable text (Wang et al., 2024c). Existing methods can be broadly categorized into two paradigms: (1) generating temporary, task-specific tools for individual queries (Qian et al., 2023), and (2) constructing reusable toolsets based on training or testing data (Cai et al., 2023; Wang et al., 2024b). These methods have demonstrated success in mathematical reasoning (Qian et al., 2023; Cai et al., 2023), visual question answering (Yuan et al., 2024; Wang et al., 2024b), and agent-based tasks (Wang et al., 2025; Zheng et al., 2025). Unlike previous works, which primarily rely on LLMs' internal knowledge, our method utilizes external references to create tools, enabling applications beyond the models' inherent knowledge scope.

**Tool-Augmented Reasoning**   Tool-augmented reasoning enhances LLMs' reasoning capabilities by integrating external tools, particularly for tasks requiring specialized knowledge or complex computation. Some studies manually curate a small set of high-quality tools (Gu et al., 2024; Lu et al., 2025), while others (Qin et al., 2024; Liu et al., 2024b) utilize large-scale APIs from platforms like RapidAPI or API-Bank (Li et al., 2023).

However, the large number of tools makes selecting the right one challenging. Prior work often relies on embedding similarity (Qin et al., 2024; Yuan et al., 2024), which may fail to capture implicit relationships when the required knowledge is not explicitly stated. In contrast, REFTOOL organizes tools within a hierarchical structure that mirrors systematic knowledge organization, enabling effective retrieval. Du et al. (2024) adopt a related strategy using RapidAPI's categorization for tool selection, whereas REFTOOL proves effective with both inherent structures in reference materials and structures constructed by LLMs.

## 6 CONCLUSION

We present REFTOOL, a framework that enhances LLM reasoning through reference-guided tool creation. Unlike prior approaches that rely solely on models' internal knowledge, REFTOOL generates code-form tools from references such as textbooks and unstructured snippets, validates them through examples, and organizes them into a hierarchical toolbox for effective selection. Experiments across causality, physics, and chemistry domains show consistent improvements over existing tool-creation and domain-specific reasoning methods, while maintaining computational efficiency. Moreover, REFTOOL generalizes beyond scientific domains, showing effectiveness in extremely low-resource language translation. By grounding tool creation and selection in authoritative references, REFTOOL

enables LLMs to go beyond the limitations of their internal knowledge, yielding accurate and broadly applicable tools. This points to a promising paradigm for extending the knowledge boundaries of LLMs and equipping them to address emerging knowledge-intensive tasks in real time.

## ACKNOWLEDGEMENTS

This work is supported by Beijing Natural Science Foundation (L253001). We thank Jiuheng Lin and Chen Zhang for suggestions on the experiments. We thank the anonymous reviewers and the area chair for their helpful suggestions.

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

## A    IMPLEMENTATION DETAILS

### A.1    EVALUATION PROTOCOL

We adopt the original benchmarks' evaluation code for consistency. The tolerant rate of numerical questions is 3% for QRData, 4% for TheoremQA, and 5% for SciBench.

### A.2    THE REFTOOL FRAMEWORK

By default, each tool's demonstration example is included during solution generation. For the causality domain, we omit the example because QRData questions involve data analysis and differ significantly in format from the examples.

The model versions are `Llama-3.1-70B-Instruct`, `gemini-1.5-pro-002`, `gpt-4-1106-preview` and `gpt-4o-2024-11-20`. The temperature of all models is set to 0. The maximum output tokens are set to 2048 for initial tool generation, refinement, and solution generation, and 512 for hierarchical tool selection. Experiments are conducted on 8 NVIDIA A800 GPUs.

### A.3    BASELINE METHODS

- **General reasoning methods**: We implement Program-of-Thoughts (PoT) for single-turn reasoning and ReAct for multi-turn reasoning. We also experiment with direct reasoning and Chain-of-Thought (Wei et al., 2022) on GPT-4 in Appendix B.1, but they are excluded from main comparisons due to inferior performance.
- **Retrieval-augmented generation (RAG) methods**: To investigate if LLMs can directly learn from reference materials, we enhance both PoT and ReAct with RAG, using the same reference books employed for tool creation. The books are segmented into subsections, and the segment with the highest similarity to the question embedding is retrieved. Reference documents are segmented by subsections. Subsections exceeding 1,000 tokens are further divided into 1,000-token chunks.

These segments are then processed using the `text-embedding-3-large` embedding model to generate text embeddings. During inference, we compute the embedding for each question and select the text with the highest similarity score to include in the model's prompt.

- **General-purpose tool creation methods**: (1) LATM (Cai et al., 2023), which crafts reusable tools for each task based on a few demonstrations; (2) Creator (Qian et al., 2023), which dynamically creates tools for each question; and (3) TroVE (Wang et al., 2024b), which uses and refines a toolbox iteratively during test time. Since LATM requires 6 instances for training and validation, we randomly sample these from the test set and evaluate on the remaining data, and all other baselines are evaluated in a zero-shot setting. For Creator, we provide a tool example from a different domain (math) in the tool creation prompt.
- **Domain-specific reasoning methods**: (1) Physics Reasoner (Pang et al., 2025), which manually constructs a formula set and instructs LLMs to retrieve formulas during reasoning; (2) StructChem (Ouyang et al., 2024), which instructs LLMs to generate formulas before reasoning; and (3) ChemAgent (Tang et al., 2025), which builds a library of memories by extensive trial-and-error on validation data. We use GPT-4o for library construction to align with REFTOOL's setting. We do not compare with web search methods as answers may be searched directly from the Internet.

### A.4  THE XLR LANGUAGE TRANSLATION EXPERIMENT

When organizing the rule structure, we allow the LLM to place one rule into multiple categories if necessary. This is because some rules cover different aspects simultaneously. For example, the rule `In Zhuang, the word "dwg" is a copular verb. In simple sentences expressing affirmative judgment, "dwg" is usually omitted. However, when expressing negation, "dwg" cannot be omitted, and the negative word "mbouj" must be placed before the copular verb "dwg."` (translated from Chinese) is placed under both categories `Word Order` and `Sentence Structure` and `Negation`. To ensure that the rules are fully unstructured in advance, we shuffle the rules before asking the LLM to organize them.

Because the rules are already acompanied with parallel examples, we only ask the LLM to create tool function in initial tool generation, and use the parallel examples in tool verification. Since all rules are collected by human experts, the tool verification is only to select tools that need further refinement, and we do not filter out any tools after refinement.

In hierarchical selection, we ask the LLM to select at most $n_c = 4$ categories and $n_t = 2$ tools under each category. On average, $3.5$ rules are selected for Chinese $\rightarrow$ Zhuang translation and $3.4$ rules for Zhuang $\rightarrow$ Chinese translation, comparable to the rule-by-rule retrieval methods.

During translation, we provide the word-to-word dictionary of the source sentence to LLMs in all methods. And each rule is accompanied with 2 parallel examples.

## B  ADDITIONAL RESULTS

### B.1  PERFORMANCE OF MORE BASELINE METHODS

Table 7: Performance of more baseline methods on GPT-4. Numbers are in percentages (%).

| Method | Accuracy | | |
|---|---|---|---|
| | Causality | Physics | Chemistry |
| Direct Reasoning | 33.1 | 14.0 | 32.8 |
| CoT | 41.3 | 10.5 | 35.4 |
| PoT | 34.2 | 45.6 | 51.8 |
| ReAct | 50.9 | 41.2 | 54.6 |

Previous works also compare with pure-text baselines like direct reasoning and Chain-of-Thought (CoT) (Wei et al., 2022), but as our preliminary experiment in Table 7 shows that these methods are much inferior to PoT on GPT-4, we do not add them into the baselines in the main paper. While CoT achieves high performance on the causality domain, this results from educated guessing on multiple-choice questions, with none of the numerical questions being answered correctly. This deviates

Table 8: Performance comparison with training Llama-3.1-70B on the reference materials (%).

| Method | Accuracy | | | |
| --- | --- | --- | --- | --- |
| | Causality | Physics | Chemistry | Average |
| PoT | 33.1 | 48.2 | 46.9 | 42.7 |
| PoT (Continued pretrained) | 34.6 | 49.1 | 48.5 | 44.1 |
| PoT (Finetuned on QA pairs) | 36.4 | 50.9 | **50.6** | 46.0 |
| PoT + RefTool | **36.8** | **53.5** | 49.5 | **46.6** |

from QRData's original goal of conducting data-based quantitative reasoning. Direct reasoning outperforms CoT in physics because, despite the instruction to answer directly, the model still generates intermediate reasoning steps for most questions.

As the code of solving physics and chemistry problems is simpler and successful executes in most cases, multi-turn reasoning is not that necessary in such scenarios, therefore we do not implement the multi-turn settings ReAct and React+REFTOOL. Table 7 also shows that ReAct introduces limited improvement or even negative influence on these domains.

We also compare RefTool with training LLMs based on the content of reference materials, with two methods: continued pretraining and fine-tuning. For continued pretraining, we use the full textbook content. For fine-tuning, we ask GPT-4o to generate question–answer pairs (QA) based on the content of each chapter. GPT-4o is asked to generate at most 10 QA pairs each time, and for causality and chemistry we run this process twice to ensure at least 1,000 training instances. We leave 10% of the data for hyperparameter search, and the training is performed for 2 epochs with the learning rate 1e-5.

As shown in Table 8, PoT + RefTool achieves the best average performance. Continued pretraining on the textbook text provides only limited improvement, which suggests that LLMs struggle to directly incorporate textbook knowledge for task solving. Fine-tuning on QA pairs narrows the performance gap, although this approach still requires transforming the textbook content through an LLM, and is more resource-intensive: Finetuning must be repeated for every target LLM, while the tools created by RefTool can be reused across models.

### B.2 Sub-dataset Performance of SciBench-chemistry

Table 9 shows the performance of sub-datasets of SciBench-chemistry. While performance varies due to each sub-dataset's small scale, REFTOOL demonstrates effectiveness in most cases.

### B.3 Performance on Reasoning Models

Table 10 shows REFTOOL's performance on o1-mini (with the specific version `o1-mini-2024-09-12`), where it improves average accuracy by 4.3% over PoT. This indicates REFTOOL's compatibility with reasoning models, effectively supplementing their knowledge and capabilities.

### B.4 Performance of Using Different Models as the Tool Creation Model

To assess the robustness of REFTOOL's tool creation module, we experiment with Gemini-1.5-Pro and Llama-3.1-70B-Instruct as alternative tool creation LLMs. As Table 11 shows, using these LLMs for tool creation also achieves superior performance compared to baseline methods. This demonstrates REFTOOL's robustness to the choice of base model and its compatibility with open-source models, which is particularly important when working with sensitive or proprietary reference materials.

Compared to GPT-4o-created tools, a relatively lower ratio of Gemini-1.5-Pro and Llama-3.1-70B-Instruct created tools passes the validation. For example, in physics, GPT-4o achieves 82% direct validation success with 8% succeeding after refinement, while Gemini-1.5-Pro achieves 54% direct success with 24% succeeding after refinement. Although refinement helps recover about one-quarter of tools, approximately 20% still get filtered out, potentially leading to incomplete knowledge coverage and slightly lower overall performance compared to GPT-4o-created tools.

Table 9: Performance of sub-datasets of SciBench-chemistry. Numbers are in percentages (%).

| Method | Accuracy | | | |
|---|---|---|---|---|
| | Chemmc | Matter | Quan | Average |
| **Llama-3.1-70B** | | | | |
| LATM | 40.6 | 23.4 | 30.3 | 31.4 |
| Creator | 50.0 | 34.0 | 36.4 | 40.1 |
| TroVE | 50.0 | 44.7 | 21.2 | 38.6 |
| StructChem | 50.0 | 21.3 | 42.4 | 37.9 |
| ChemAgent | 60.5 | 44.7 | 39.4 | 48.2 |
| PoT | 65.8 | 44.7 | 30.3 | 46.9 |
| PoT + RAG | 63.2 | 44.7 | 36.4 | 48.1 |
| PoT + REFTOOL | 63.2 | 48.9 | 36.4 | 49.5 |
| **Gemini-1.5-Pro** | | | | |
| LATM | 28.1 | 17.0 | 30.3 | 25.1 |
| Creator | 73.7 | 63.8 | 42.4 | 60.0 |
| TroVE | 81.6 | 63.8 | 51.5 | 65.6 |
| StructChem | 57.9 | 38.3 | 54.5 | 50.2 |
| ChemAgent | 78.9 | 66.0 | 51.5 | 65.5 |
| PoT | 78.9 | 59.6 | 48.5 | 62.3 |
| PoT + RAG | 78.9 | 63.8 | 48.5 | 63.7 |
| PoT + REFTOOL | 81.6 | 66.0 | 51.5 | 66.4 |
| **GPT-4** | | | | |
| LATM | 53.1 | 42.6 | 39.4 | 45.0 |
| Creator | 60.5 | 46.8 | 33.3 | 46.9 |
| TroVE | 55.3 | 31.9 | 30.3 | 39.2 |
| StructChem | 36.8 | 19.1 | 33.3 | 29.7 |
| ChemAgent | 68.4 | 46.8 | 42.4 | 52.5 |
| PoT | 68.4 | 44.7 | 42.4 | 51.8 |
| PoT + RAG | 65.8 | 51.1 | 45.5 | 54.1 |
| PoT + REFTOOL | 71.1 | 46.8 | 42.4 | 53.4 |
| **GPT-4o** | | | | |
| LATM | 43.8 | 36.2 | 27.3 | 35.7 |
| Creator | 55.3 | 38.3 | 36.4 | 43.3 |
| TroVE | 71.1 | 44.7 | 42.4 | 52.7 |
| StructChem | 55.3 | 29.8 | 36.4 | 40.5 |
| ChemAgent | 78.9 | 55.3 | 42.4 | 58.9 |
| PoT | 78.9 | 55.3 | 42.4 | 58.9 |
| PoT + RAG | 76.3 | 51.1 | 42.4 | 56.6 |
| PoT + REFTOOL | 76.3 | 53.2 | 54.5 | 61.3 |

Table 10: Performance of o1-mini. Numbers are in percentages (%), with the best performance for each model shown in bold.

| Method | Accuracy | | |
|---|---|---|---|
| | Causality | Physics | Chemistry |
| PoT | 44.2 | 56.1 | 60.5 |
| PoT + REFTOOL | **50.2** | **57.9** | **65.6** |

## B.5 PERFORMANCE ON ANOTHER PHYSICS DATASET: SCIBENCH-FUND

We evaluated REFTOOL on another physics dataset SciBench-fund (Wang et al., 2024a) with 71 questions, to test tool generalizability.[7] Table 12 shows that REFTOOL outperforms all zero-shot baselines and matches 4-shot Physics Reasoner's performance using the same tools in the evaluation of TheoremQA. This demonstrates REFTOOL's dataset-agnostic nature, where domain-specific tools can be applied across different datasets.

---

[7]We excluded two other SciBench-physics sub-datasets as they require advanced thermodynamics and particle dynamics knowledge beyond our reference textbook's scope.

Table 11: Performance of REFTOOL using Gemini-1.5-Pro and Llama-3.1-70B as the tool creation model. Numbers are in percentages (%), with the best performance for each model shown in bold.

| Method | Accuracy | | | | |
| --- | --- | --- | --- | --- | --- |
| | Llama-3.1-70B | Gemini-1.5-Pro | GPT-4 | GPT-4o | Average |
| **Physics** | | | | | |
| Physics Reasoner | 48.2 | 50.9 | 42.1 | 33.3 | 43.4 |
| LATM | 38.9 | 33.3 | 39.8 | 30.6 | 35.7 |
| Creator | 40.4 | 57.0 | 35.1 | 40.4 | 43.2 |
| TroVE | 33.3 | 58.8 | 35.1 | 48.2 | 43.9 |
| PoT | 48.2 | 57.9 | 45.6 | 57.0 | 52.2 |
| PoT + RAG | 44.7 | 57.0 | 44.7 | 57.9 | 51.1 |
| PoT + REFTOOL (GPT-4O) | **53.5** | 58.8 | 49.1 | 57.9 | **54.8** |
| PoT + REFTOOL (GEMINI) | 48.2 | 58.8 | 47.4 | **58.8** | 53.3 |
| PoT + REFTOOL (LLAMA) | 49.1 | **60.5** | **50.0** | 57.0 | 54.2 |
| **Chemistry** | | | | | |
| LATM | 31.4 | 25.1 | 45.0 | 35.7 | 34.3 |
| Creator | 40.1 | 60.0 | 46.9 | 43.3 | 47.6 |
| TroVE | 38.6 | 65.6 | 39.2 | 52.7 | 49.0 |
| StructChem | 37.9 | 50.2 | 29.7 | 40.5 | 39.6 |
| ChemAgent | 48.2 | 65.5 | 52.5 | 58.9 | 56.3 |
| PoT | 46.9 | 62.3 | 51.8 | 58.9 | 55.0 |
| PoT + RAG | 48.1 | 63.7 | **53.7** | 56.6 | 55.5 |
| PoT + REFTOOL (GPT-4O) | **49.5** | **66.4** | 53.4 | 61.3 | **57.7** |
| PoT + REFTOOL (GEMINI) | 48.3 | 64.9 | 52.5 | **61.6** | 56.8 |
| PoT + REFTOOL (LLAMA) | **49.5** | 62.4 | 53.2 | 57.3 | 55.6 |

Table 12: Performance on another physics dataset: Scibench-fund. Numbers are in percentages (%), with the best performance for each model shown in bold.

| Method | Accuracy | | | | |
| --- | --- | --- | --- | --- | --- |
| | Llama-3.1-70B | Gemini-1.5-Pro | GPT-4 | GPT-4o | Average |
| Physics Reasoner | 56.3 | 59.2 | 63.4 | 36.6 | 53.9 |
| Creator | 56.3 | 70.4 | 53.5 | 57.7 | 59.5 |
| PoT | 53.5 | 69.0 | 59.2 | 73.2 | 63.7 |
| PoT + RAG | 54.9 | **73.2** | 59.2 | 73.2 | 65.1 |
| PoT + REFTOOL | **57.7** | **73.2** | **64.8** | **74.6** | **67.6** |
| Physics Reasoner (4-shot) | 62.0 | 73.2 | 63.4 | 71.8 | 67.6 |

## B.6 COST ANALYSIS

Table 13 shows that REFTOOL is highly efficient during inference, compared with all the tool creation and domain-specific reasoning baseline methods. Only LATM costs less than REFTOOL, but its performance is much inferior to REFTOOL. Even when including tool creation costs, REFTOOL remains more efficient than most tool-augmented methods. Furthermore, because the tools are reusable, the creation cost is amortized and remains fixed regardless of the number of inference instances.

## C HUMAN EVALUATION: CONSISTENCY OF TOOL SELECTION WITH HUMANS

We compare human and LLM tool selection by having experts simulate the hierarchical selection process. For each domain, we randomly sample 20 questions where models consistently use tools.

In the tool selection process, given a question, annotators are first asked to select at most one category from the given book, and none if no category is relevant to the question. If they select a category, they are then asked to select one tool within the category if it is the most useful, select two tools only if they are equally useful, and select none if none of the tools are useful. For each domain, we

Table 13: Cost analysis of tool-augmented and domain-specific methods (with GPT-4o as the base model). "Human" indicates that the step is done by humans and the cost is unknown. As Creator and Trove create tools during inference, they do not have a seperate toolbox construction cost.

| Domain | Method | Time (min.) | | Cost ($) | |
|---|---|---|---|---|---|
| | | Toolbox Construction | Inference | Toolbox Construction | Inference |
| Physics | LATM | 1 | 4 | 0.1 | 1.2 |
| | Creator | - | 59 | - | 3.3 |
| | TroVE | - | 22 | - | 1.6 |
| | Physics Reasoner | Human | 75 | Human | 3.5 |
| | PoT + REFTOOL | 5 | 2 | 6.9 | 1.5 |
| Chemistry | LATM | 1 | 4 | 0.1 | 1.1 |
| | Creator | - | 49 | - | 2.6 |
| | TroVE | - | 52 | - | 7.5 |
| | StructChem | - | 142 | - | 8.7 |
| | ChemAgent | 1233 | 536 | 79.3 | 41.3 |
| | PoT + REFTOOL | 3 | 6 | 3.5 | 1.4 |

Table 14: Consistency of tool selection with humans (%). Category selection consistency is calculated as the fraction of questions where human and model select the same category. Tool selection consistency is the fraction where their tools overlap, given they both choose tools from the same category.

| Domain | Consistency | Llama-3.1-70B | Gemini-1.5-Pro | GPT-4 | GPT-4o |
|---|---|---|---|---|---|
| Causality | Category Selection | 100 | 95 | 100 | 100 |
| | Tool Selection within Category | 94 | 100 | 91 | 94 |
| Physics | Category Selection | 80 | 80 | 75 | 76 |
| | Tool Selection within Category | 53 | 56 | 44 | 69 |
| Chemistry | Category Selection | 55 | 65 | 72 | 75 |
| | Tool Selection within Category | 60 | 67 | 40 | 90 |

randomly sample 20 questions where most models (at least 3 out of 4) choose to use tools. All human annotators are fairly paid.

**Consistency Metrics** For category selection, the consistency is computed as:

$$\text{Consistency}_{\text{category}} = \frac{|\{\text{human and model select the same category}\}|}{|\{\text{both human and model select a category}\}|}.$$

And for tool selection, the consistency is computed as

$$\text{Consistency}_{\text{tool}} = \frac{|\{\text{overlap exists between tools selected by human and model}\}|}{|\{\text{both human and model select tools within the same category}\}|}.$$

Table 14 shows the consistency between LLMs and human experts. Agreement is higher in category selection than in tool selection, supporting our hierarchical selection step which narrows down the tool search space with a high consensus. Across domains, causality shows the strongest consistency in both category and tool selection, owing to more direct questions with keywords like *average treatment effect* that clearly indicate the relevant knowledge. In contrast, physics and chemistry questions often involve indirect formulations that make it harder for models to identify the required knowledge.

Gemini-1.5-Pro and GPT-4o demonstrate better alignment with human experts, mirroring their superior PoT performance which reflects stronger internal domain knowledge. While Llama-3.1-70B and GPT-4 show weaker consistency with humans in physics and chemistry tool selection, their chosen tools are still valuable as relevant knowledge is recalled. In cases where these models select the same category as humans but different tools, we observe a 17% accuracy improvement from tool usage compared to the PoT baseline. Appendix D provides a concrete example.

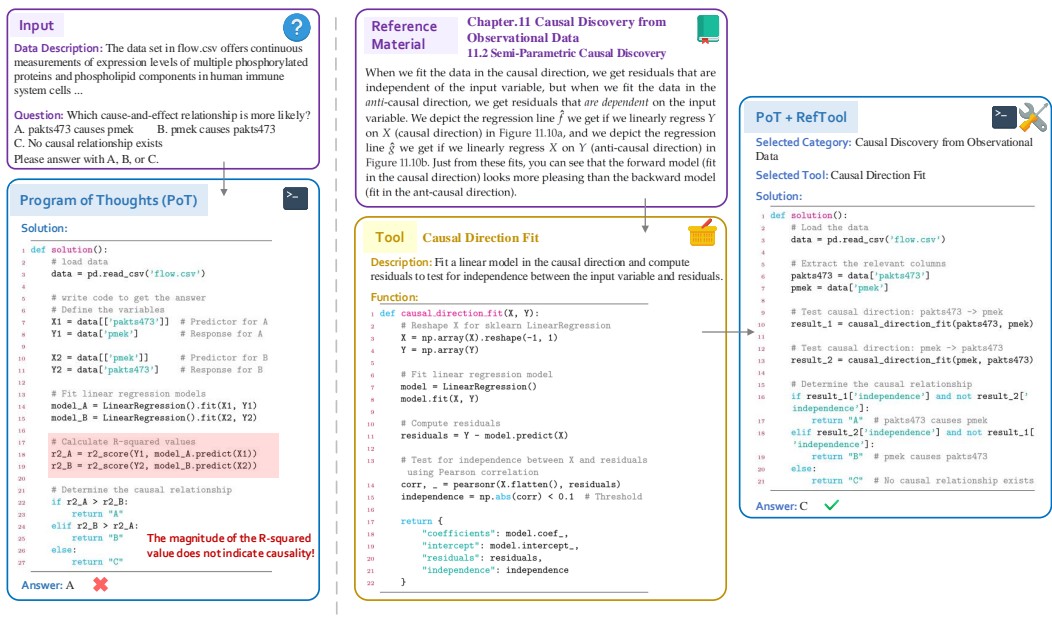

Figure 4: Example case of GPT-4o on a causal problem with (right) and without (left) REFTOOL. This is the detailed version of Figure 3

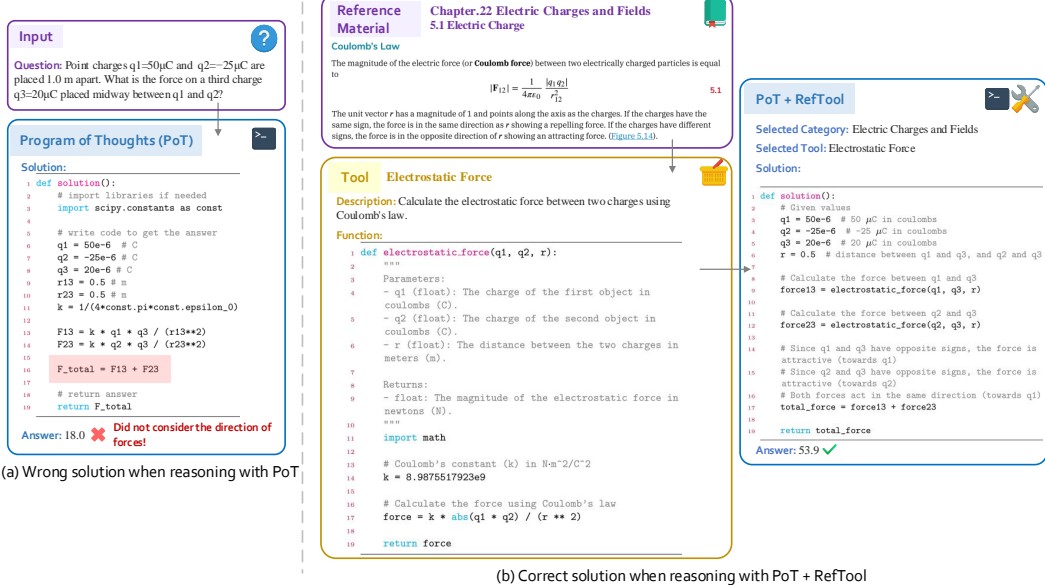

Figure 5: Example case of Gemini-1.5-Pro on a physical problem with (right) and without (left) REFTOOL.

# D CASE STUDY

Figure 4 provides the detailed causality case discussed in §4.4, while Figures 5 and 6 show physics and chemistry cases. These cases illustrate how REFTOOL helps LLMs solve problems when standard PoT fails. Notably, in Figure 6, while the selected tool doesn't directly solve the question, it provides relevant knowledge for the LLM to solve the question in a roundabout way (through the ionization energy of hydrogen), which is different from the expert solution but also leads to the correct answer.

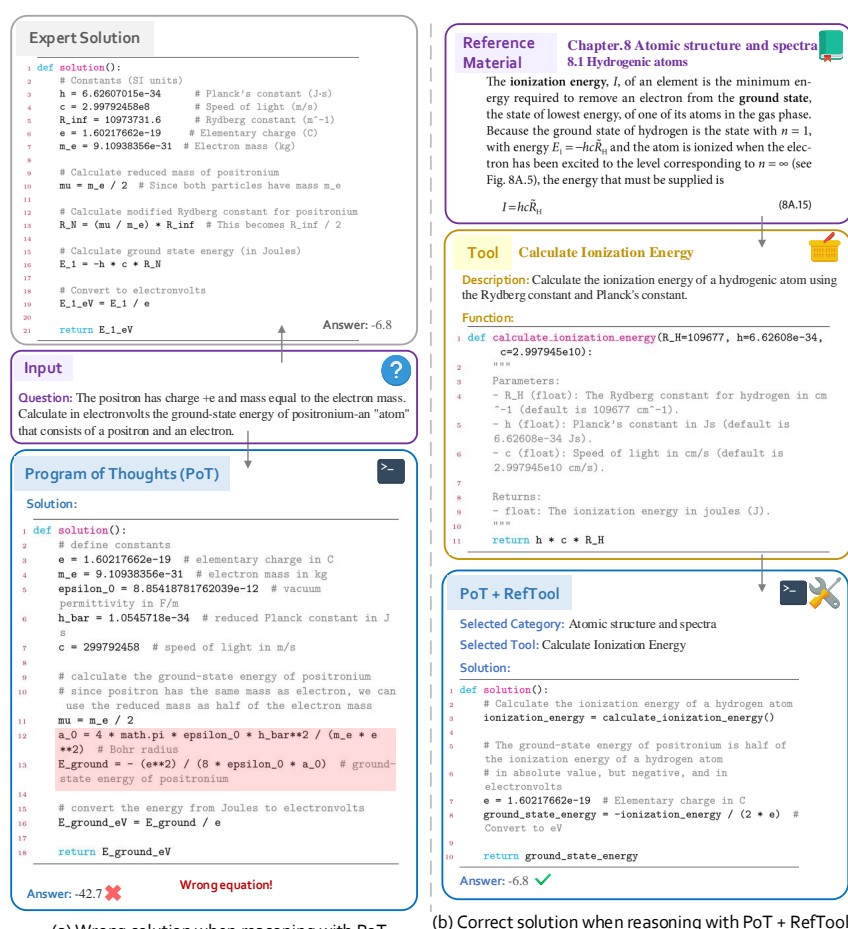

Figure 6: Example case of Llama-3.1-70b on a chemical problem with (right) and without (left) REFTOOL.

# E    THE USE OF LLMS

We use LLMs in polishing the writing, but LLMs do not play a significant role like research ideation or writing the content directly.

# F    PROMPTS

Figure 7 - Figure 11 demonstrate the prompts of REFTOOL.

Prompts of the general-reasoning baselines as shown in Figure 12 and Figure 13 are designed with reference to the QRData and SciBench papers. Prompts for ReAct are the same as the QRData paper.

---

**Prompt for Category Generation**

Here are rules related to {task}. Your task is to identify top-level categories for organizing these rules.
Read through all the rules and determine the major branches of {task} they represent. Ideally there should be about 10-15 top-level categories. One rule can be in multiple categories if necessary (at most two).
Output only the names of these categories as a JSON list.
{segments}

---

**Prompt for Assigning Reference Segments to Categories**

Here is a rule related to {task} and the category names that have been identified:
Categories:
{categories}
Rule: {rule}
Your task is to classify this rule into at most two categories and create a descriptive name for this rule under each category.
Output in JSON format where:
- The keys are the category names that this rule belongs to
- The value is the name of the rule under this category (a concise, descriptive name)
Example output:
{{
"Category 1": "Descriptive rule name for Category 1",
"Category 2": "Descriptive rule name for Category 2"
}}
Note: The rule should be classified into 1-2 categories maximum.

---

Figure 7: Prompt template for knowledge organization.

---

**Prompt for Initial Tool Generation**

Please extract the skills from the following text. The text is a section from the chapter {chapter} of the book {book}.
Each skill is a python function with comments of parameters and returns, accompanied by a description and a demonstration example of using the skill.
Please limit the number of skills to 2, and organize the skills in a list of json objects.
Please implement the function, and *do not* leave it as a placeholder. Note the indent in code is 4 spaces. All packages used should be imported inside the function. The function should be self-contained.
If the text contains examples, you are encouraged to use the examples in the text, otherwise please design examples by yourself. The answer to the example question is encouraged to be numerical.

---

NOTE THAT THE SKILL PYTHON CODE SHOULD NOT BE SPECIFIC TO/ONLY APPLIED TO THE CHOSEN EXAMPLE! PLEASE GENERATE GENERAL SKILL CODE.
The output should be in *complete* json structure, starting with '[' and ending with ']'.

Example output:
[{
"description": "Compute the expected return using the Capital Asset Pricing Model (CAPM) formula.",
"function": """def expected_return(rf, beta, rm):
\"\"\"
Parameters:
- rf (float): The risk-free rate.
- beta (float): The beta of the portfolio.
- rm (float): The return on the market.
Returns:
- float: The expected return.
\"\"\"
return rf + beta * (rm - rf)""",
"example": {
"question": "Suppose a stock has the following information. It is listed on the London stock exchange and operates throughout Europe. The yield on a UK 10 year treasury is 2.8%. The stock in question will earn 8.6% as per historical data. The Beta for the stock is 1.4, i.e., it is 140% volatile to the changes in the general stock market. What is the expected rate of return?",
"solution": """def solution():
# Given values.
rf = 0.028 # The yield on a UK 10 year treasury
beta = 1.4 # The stock is 140% volatile to the changes in the general stock market
rm = 0.086 # The stock in question will earn 8.6% as per historical data
# Calculate the expected return .
result = expected_return(rf, beta, rm)
# Return the result.
return result""",
"answer": 0.109
}}]

Text:
{text}

Figure 8: Prompt template for initial tool generation.

**Prompt for Tool Refinement**

Please revise the skill according to the feedback.
The skill is a python function with comments of parameters and returns, accompanied by a description and a demonstration example of using the skill. Please try to keep the original intent of the skill, and modify the description/function/example to address the feedback.
Note the indent in code is 4 spaces. All packages used should be imported inside the function. The function should be self-contained. The answer to the example question is encouraged to be numerical.
NOTE THAT THE SKILL PYTHON CODE SHOULD NOT BE SPECIFIC TO/ONLY APPLIED TO THE CHOSEN EXAMPLE! PLEASE GENERATE GENERAL SKILL CODE.
The output should be in *complete* json structure as the original skill, starting with '{' and

ending with '}'.

Original Skill:
{skill}

Feedback:
{feedback}

Figure 9: Prompt template for tool refinement.

**Prompt for Category Selection**

You are a data analyst and good at quantitative reasoning. You are required to respond to a quantitative question using the provided data.
The question can be found below. Given the table of content of the book {book}, please select the chapters that you find useful in solving the question.
Please provide an explanation supporting your choice. At the last line of your response, format the number of the chapters with a list, like '[0]'. Limit the number of chapters to at most 1. Output '[]' if none of the chapters are useful. The last line should start with '[' and end with ']'.

Question:
{question}

Table of Content:
{table_of_content}

Response:

**Prompt for Tool Selection within Category**

You are a data analyst and good at quantitative reasoning. You are required to respond to a quantitative question.
The question and the list of skills can be found below. Please select the skills that you find useful in solving the question
Please provide an explanation supporting your choice. At the last line of your response, format the number of the skills with a list, like '[0]'. Limit the number of skills to at most 2. Output '[]' if none of the skills are useful. The last line should start with '[' and end with ']'.

Question:
{question}

List of skills:
{tools}

Response:

Figure 10: Prompt template for tool selection.

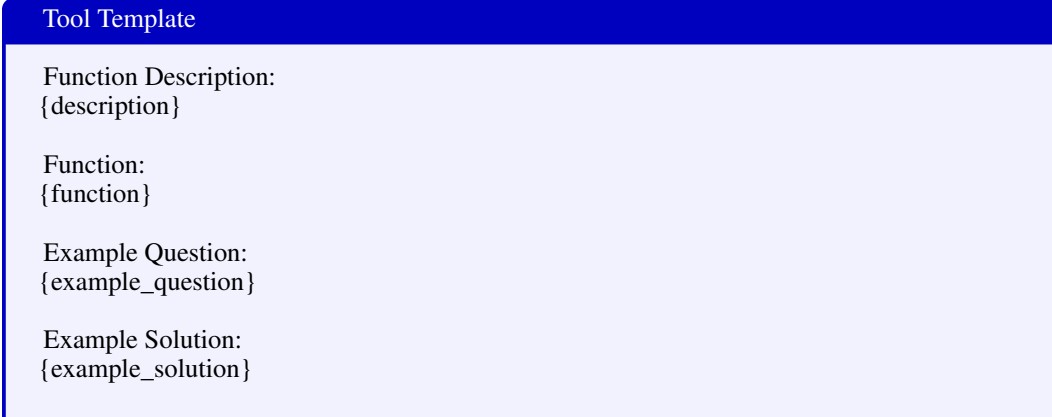

**Prompt for Solution Generation**

You are a data analyst and good at quantitative reasoning. You are required to respond to a quantitative question below. Please write python code to answer the question. Please encase the Python code within triple backticks. You can use any python library you imported. The returned value of the code is supposed to be the answer. The format of the code should be
```python
def solution():
# import libraries if needed

# write code to get the answer

# return answer
```

Question:
{question}

Please note that we provide you several functions for the above question. If the functions are related to the question, you are encouraged to use the functions to solve the question. The functions will also be provided in execution, so just call them. *DO NOT* define the functions again or import the functions.

Functions:
{tools}

Response:

**Tool Template**

Function Description:
{description}

Function:
{function}

Example Question:
{example_question}

Example Solution:
{example_solution}

Figure 11: Prompt template for solution generation. For evaluation of QRData, the data description and ten lines of the shuffled data are also added to the prompt along with the question.

Prompt for PoT

You are a data analyst and good at quantitative reasoning. You are required to respond to a quantitative question below. Please write python code to answer the question. Please encase the Python code within triple backticks. You can use any python library you imported. The returned value of the code is supposed to be the answer. The format of the code should be
```python
def solution():
# import libraries if needed

# write code to get the answer

# return answer
```

Question:
{question}

Response:

Figure 12: Prompt template for PoT. For evaluation of QRData, the data description and ten lines of the shuffled data are also added to the prompt along with the question.

Prompt for CoT

You are a data analyst and good at quantitative reasoning. You are required to respond to a quantitative question below. Please provide a clear and step-by-step solution to answer the question. Do not write any code in your answer. Conclude the answer by stating "The answer is therefore \boxed{[ANSWER]}."

Question:
{question}

Response:

Prompt for Direct Reasoning

You are a data analyst and good at quantitative reasoning. You are required to respond to a quantitative question below. Directly answer by stating "The answer is therefore \boxed{[ANSWER]}."

Question:
{question}

Response:

Figure 13: Prompt templates for CoT and direct reasoning. For evaluation of QRData, the content of the data (shuffled and truncated to the first 3500 tokens) is also added to the prompt along with the question. For evaluation of SciBench, the prompt also states "The question will specify the unit of measurement, which should not be included in the answer. Express the final answer as a decimal number with three digits after the decimal point."

