# OpenReview forum: "RefTool: Reference-Guided Tool Creation for Knowledge-Intensive Reasoning"
_ICLR.cc/2026/Conference — ICLR 2026 Poster_

### Official Review · Reviewer_PQMZ · 2025-10-29

**Soundness:** 4
**Presentation:** 3
**Contribution:** 3
**Rating:** 6
**Confidence:** 3

**Summary:**

This paper introduces REFTOOL, a framework for tool-augmented reasoning designed to enhance the inferential capabilities of  LLMs in knowledge-intensive domains that lack predefined tools. REFTOOL leverages external materials to perform hierarchical tool creation, operating through two primary modules:

Tool Creation: The LLM processes reference content to generate executable tools. Each tool is generated with a description, its function code, and an illustrative example. These tools are subsequently validated using their own examples and optimized as necessary. Finally, the validated tools are organized into a hierarchical toolbox that generally mirrors the structure of the reference material.

Tool Utilization: During the reasoning process, when a query is encountered, the LLM executes a hierarchical selection procedure. It will select the most relevant category and subsequently selects the most appropriate tool from within that category. The selected tool is then employed to generate the final solution.

Experiments were conducted on causality, physics, and chemistry benchmarks. The results demonstrate that REFTOOL's mean accuracy significantly outperforms existing tool-creation methods, domain-specific models, and RAG-based approaches by an average of 12.3%. The paper also shows that the framework is cost-effective and exhibits high generalizability, having been successfully applied to non-scientific tasks using unstructured reference materials.

**Strengths:**

1. Novel Framework for Knowledge Proceduralization: The paper introduces a novel paradigm that bridges declarative knowledge retrieval with executable tool use. By compiling procedural knowledge from static reference materials into a persistent, validated, and structured toolbox, the framework (REFTOOL) presents a significant conceptual advance over standard Retrieval-Augmented Generation (RAG) and pre-defined tool-use methodologies.

2. Robust Methodological Design and Validation: The framework's two-module architecture is methodologically sound, incorporating critical components for robustness, such as automated tool verification, refinement loops , and a hierarchical selection mechanism. The efficacy of these design choices is rigorously validated through comprehensive ablation studies, which demonstrate the clear superiority of code-based tools over raw text retrieval and hierarchical selection over standard similarity-based retrieval.

3. Rigorous Experiments and Generalizability Validation:  The effectiveness of REFTOOL is substantiated through extensive experiments across three distinct, knowledge-intensive scientific domains, showing consistent and significant performance gains over a wide array of strong baselines. Furthermore, the framework's generalizability is impressively demonstrated by its successful application to a non-scientific task using unstructured reference materials, highlighting its flexibility and broad applicability.

**Weaknesses:**

1. Dependency on the Capability of the Base Model for Tool Creation: The core function of the REFTOOL framework relies on a specific base model to generate high-quality and validated tools. However, the framework's performance varies significantly under different base models. Supplemental experiments in Appendix B.4 show that other models, such as Gemini-1.5-Pro and Llama-3.1-70B, have much lower initial validation success rates. For example, in the physics domain, the success rate was 54% versus 82%.

2. Insufficient handling of prerequisite foundational knowledge. The design of the framework implicitly assumes that the knowledge in reference materials is entirely comprehensible. However, in specialized fields such as chemistry, textbooks often presuppose that readers have mastered undefined foundational knowledge (e.g., IUPAC nomenclature, SMILES notation). If the tool creation model misinterprets these prerequisite foundational concepts, the model may fail to generate correct tools.

**Questions:**

This paper's core premise is that REFTOOL can overcome an LLM's internal knowledge limitations by leveraging reference materials. However, in specialized domains like chemistry, reference materials often assume a baseline of foundational knowledge rather than explicitly teaching it. For instance, a textbook may freely use representations like SMILES strings or IUPAC nomenclature in its examples and formulas, assuming the reader is already proficient.

During Tool Creation: If the base LLM lacks reliable pre-trained knowledge of (e.g.) SMILES, it may fundamentally misinterpret a textbook example and thus generate a semantically flawed tool. The Tool Verification step, which relies on a model-generated example, could also fail to catch this error if the model produces a self-consistent but incorrect validation case, leading to the propagation of a bad tool.

During Tool Utilization: Even if a tool is perfectly created (e.g., calculate_property(smiles_string)), it becomes practically useless if the LLM, when given a natural language query ("calculate the boiling point of ethanol"), cannot perform the implicit prerequisite task of translating "ethanol" into its required SMILES format ("CCO"). This adds an additional reasoning burden that the framework was intended to solve, not create.

Does the REFTOOL architecture possess a specific design or mechanism to address this gap in implicit foundational knowledge?

For example, does the framework (a) simply rely on the comprehensiveness of the reference material to also teach these fundamentals, or (b) does it include a more active mechanism to detect these dependencies and recursively build a foundational library of prerequisite tools (e.g., a chemical_name_to_smiles converter) that the higher-level scientific tools can then rely upon?

---

> ### Author Response · Authors · 2025-11-20
> **Response to Reviewer PQMZ (1/2)**
>
> Thank you for your constructive comments! Below we address your concerns in detail.
>
> 1. Dependency on the capability of the tool creation model
>
> > The core function of the REFTOOL framework relies on a specific base model to generate high-quality and validated tools. However, the framework's performance varies significantly under different base models. Supplemental experiments in Appendix B.4 show that other models, such as Gemini-1.5-Pro and Llama-3.1-70B, have much lower initial validation success rates. For example, in the physics domain, the success rate was 54% versus 82%.
>
> To validate the quality of tools generated by Gemini-1.5-Pro, we conduct a human evaluation of 20 tools for each domain, and the results are shown below.
>
> | **Domain** | **Faithful** | **Function Correct** | **Example Correct** | **Useful** |
> |-----------------|-------------------|---------------------------|--------------------------|-----------------|
> | Causality       |95                | 95                        | 100                     | 85              |
> | Physics         | 95               | 95                        | 100                      | 95              |
> | Chemistry       | 90                | 80                        | 100                       | 85              |
>
> Although only 54% tools pass the initial validation, the final tools after refinement are of decent quality, although Gemini encounters greater difficulty in generating correct chemical tools. Experiments of using Gemini and Llama created tools, as shown in Table 10 of the paper, show that these tools also improve downstream reasoning performance. This demonstrates that RefTool remains helpful even when the base model for tool creation is weaker.
>
> 2. Foundational knowledge required for tool creation
>
> > The design of the framework implicitly assumes that the knowledge in reference materials is entirely comprehensible. However, in specialized fields such as chemistry, textbooks often presuppose that readers have mastered undefined foundational knowledge (e.g., IUPAC nomenclature, SMILES notation). If the tool creation model misinterprets these prerequisite foundational concepts, the model may fail to generate correct tools.
>
> > The Tool Verification step, which relies on a model-generated example, could also fail to catch this error if the model produces a self-consistent but incorrect validation case, leading to the propagation of a bad tool.
> In our current experiments, the reference materials fall within the understanding capability of the tool creation LLMs, as supported by the human evaluation results. We also expanded the human evaluation in Table 5 to 50 tools per domain, making the assessment more reliable.
>
> To obtain higher quality validation examples, we encourage the tool creation LLM to reuse example problems present in the reference text whenever possible. Manual inspection of 20 chemistry tools shows that half of them include examples directly taken from the textbook with identical solutions, and another quarter use the same numerical conditions while posing a modified question. This supports the reliability of validation based on examples.
>
> For more specialized fields that require significant background knowledge, additional procedures can further enhance tool quality. These include fine-tuning the tool creation LLM, enabling the model to search for supplementary prerequisite knowledge, or incorporating interactive human expert feedback. Such approaches can help ensure that the model fully understands the reference materials before creating tools.

---

> ### Author Response · Authors · 2025-11-20
> **Response to Reviewer PQMZ (2/2)**
>
> 3. Handling prerequisite knowledge in tool utilization
>
> > During Tool Utilization: Even if a tool is perfectly created (e.g.,calculate_property(smiles_string)), it becomes practically useless if the LLM, when given a natural language query ("calculate the boiling point of ethanol"), cannot perform the implicit prerequisite task of translating "ethanol" into its required SMILES format ("CCO"). This adds an additional reasoning burden that the framework was intended to solve, not create.
>
> > Does the REFTOOL architecture possess a specific design or mechanism to address this gap in implicit foundational knowledge?
>
> Thank you for this insightful comment. For less advanced LLMs, it is indeed valuable to construct a library of prerequisite tools. Such tools can be created using more basic reference materials, for example a general chemistry textbook, or can be generated directly by instructing the tool creation LLM to identify and create prerequisite tools that support the existing toolset.
>
> We experiment with the second option. After GPT-4o generates tools for the chemistry textbook, we provide the tools from each chapter back to GPT-4o and ask it to create prerequisite tools that facilitate their use. This results in 87 tools, of which 81 pass validation and revision. Most of these tools support unit conversion or physical constant lookup. We equip two smaller LLMs, Llama-3.1-8B and Qwen2.5-7B, with these prerequisite tools. The results are shown below.
>
> |  | **Llama-3.1-8b** | **Qwen2.5-7b** |
> |--|--|--|
> | PoT | 21.1 | 23.8 |
> | PoT + RefTool (original) | 20.4 | 24.9 |
> | PoT + RefTool (w. prerequisite tools) | **24.6** | **26.6** |
>
> The original RefTool toolset does not significantly benefit these smaller models and even causes a performance drop in Llama-3.1-8B. Adding the prerequisite tools leads to a larger performance gain for both models. This shows that supporting foundational knowledge is important for enabling less capable LLMs to effectively use more advanced domain tools. We will incorporate this analysis and the associated experiment into the paper.

---

> ### Comment · Reviewer_PQMZ · 2025-11-26
>
> I appreciate the authors for their detailed responses and the additional experiments conducted during the rebuttal phase.
>
> Regarding the issue of model dependency: Although the human evaluation of tools generated by Gemini-1.5-Pro indicates that the final tools are usable, the relatively low validation success rate (54%) compared to GPT-4o demonstrates that the robustness and generalizability claimed by the framework are limited when utilizing less capable models.
>
> Regarding the issue of implicit prerequisite knowledge: The experiments confirmed the effectiveness of generating prerequisite tools, but they also exposed the original framework's inability to handle implicit knowledge: without these additional interventions, the core method cannot be fully self-sufficient in specialized domains.
>
> Conclusion: This rebuttal addressed my questions, but it also indicates that the issues I pointed out do indeed exist to some extent. The results of the supplementary experiments explored the capability boundaries of the framework but did not fundamentally resolve the aforementioned issues. I will maintain my original score.

---

> > ### Author Response · Authors · 2025-11-26
> >
> > Thank you for the thoughtful follow-up! We appreciate that you carefully reviewed our rebuttal, and we would like to offer a few clarifications:
> >
> > 1. Model dependency:
> >
> > We agree that the capability of the tool-creation model influences the quality of the generated tools. However, we would like to emphasize two observations:
> > (1) The verification and refinement stage is effective at filtering out incorrect tools, resulting in high-quality final tools across models;
> > (2) Even tools created by less capable models still yield clear performance improvements during inference.
> > Together, these findings suggest that our framework is robust, even when the underlying model is not among the strongest.
> >
> > 2. Implicit prerequisite knowledge
> >
> > The prerequisite tools indeed help improve performance for smaller or less capable models such as Llama-3.1-8B and Qwen2.5-7B, specifically in cases where *the model lacks essential domain knowledge to use the tools effectively*. However, when we provide the same prerequisite tools to the stronger models used in our main experiments (e.g., Llama-3.1-70B and GPT-4o), their performance does not improve further.
> >
> > This suggests that prerequisite-tool generation could work as an optional, situation-specific module rather than a core requirement of the framework. We appreciate your suggestion to incorporate this module, and we will discuss this in the revised paper.

---

### Official Review · Reviewer_m8Qh · 2025-10-31

**Soundness:** 2
**Presentation:** 3
**Contribution:** 2
**Rating:** 4
**Confidence:** 4

**Summary:**

This paper presents REFTOOL, a framework that automatically builds and uses code-based tools for knowledge-intensive reasoning. It works in two stages: (1) tool creation, where an LLM reads reference materials, generates and validates functions, and organizes them into a hierarchical toolbox; and (2) tool use, where the LLM selects and applies these tools to solve new problems. Experiments in causality, physics, and chemistry show that REFTOOL outperforms existing tool-creation and domain-specific methods, and it also generalizes well to a low-resource language translation task.

**Strengths:**

1. Principled approach to grounding tool generation in external, authoritative sources. It is a well-motivated and logical solution for specialized domains.
2. The introduction of a hierarchical toolbox structure that mirrors the organization of the reference material is a strong design choice.
3. Demonstrated generalization.

**Weaknesses:**

1. The framework's performance depends on the quality and structure of the reference material. It assumes a well-organized, comprehensive textbook with content that can be easily split into discrete functions. How does the framework cope with noisy, theoretical, or poorly structured references?
2. The core components are incremental and miss the core baseline, such as fine-tuning the base LLM directly on textbook content. It is unclear whether the explicit, complex, and potentially brittle tool-creation pipeline is truly superior to simply teaching the model the domain knowledge through continued pre-training or fine-tuning.
3. Human evaluation of tool quality uses a small sample—just 20 tools per domain—making it hard to generalize across hundreds created. While REFTOOL outperforms baselines, the gains are often modest. For example, chemistry scores improve by only 2% over POT + RAG (Table 2), questioning the complexity of the pipeline. Also, Table 5's cost analysis omits the considerable manual and computational effort required for reference curation and initial tool generation.

**Questions:**

The hierarchical selection is a critical component. Could you provide metrics on the accuracy of the category and tool selection steps themselves, perhaps against an oracle or human judgment? What is the failure rate at these stages?

---

> ### Author Response · Authors · 2025-11-20
> **Response to Reviewer m8Qh (1/2)**
>
> Thank you for your helpful comments! Below we address your concerns in detail.
>
> 1. Handling unstructured reference material
>
> > The framework's performance depends on the quality and structure of the reference material. It assumes a well-organized, comprehensive textbook with content that can be easily split into discrete functions. How does the framework cope with noisy, theoretical, or poorly structured references?
>
> When the reference material does not contain an inherent structure, we instruct the tool creation LLM to first propose a hierarchical structure. The experiment on extremely low-resource language translation in Section 3.3 illustrates this process. Despite having limited prior knowledge of the target language Zhuang, the LLM uses its general linguistic understanding to introduce categories such as *Numerals and Quantifiers* and *Word Order and Sentence Structure*.
>
> Moreover, the reference material consists of descriptive grammar rules rather than functions, and the tool creation LLM converts these descriptions into pseudo-code style tools. These tools help the model understand and apply the rules, leading to better translation performance.
>
> 2. Comparison with training baselines
>
> > The core components are incremental and miss the core baseline, such as fine-tuning the base LLM directly on textbook content. It is unclear whether the explicit, complex, and potentially brittle tool-creation pipeline is truly superior to simply teaching the model the domain knowledge through continued pre-training or fine-tuning.
>
> Thank you for the suggestion! We conduct continued pretraining and fine-tuning experiments on Llama-3.1-70B. For continued pretraining, we use the full textbook content. For fine-tuning, we ask GPT-4o to generate question–answer pairs (QA) based on the content of each chapter. GPT-4o is asked to generate at most 10 QA pairs each time, and for causality and chemistry we run this process twice to ensure at least 1,000 training instances. We leave 10% of the data for hyperparameter search, and the training is performed for 2 epochs with the learning rate 1e-5.
>
> |  | **Causality** | **Physics** | **Chemistry** | **Average** |
> |--|--|--|--|--|
> | PoT | 33.1 | 48.2 | 46.9 | 42.7 |
> | PoT (Continued pretrained) | 34.6 | 49.1 | 48.5 | 44.1 |
> | PoT (Finetuned on QA pairs) | 36.4 | 50.9 | **50.6** | 46.0 |
> | PoT + RefTool | **36.8** | **53.5** | 49.5 | **46.6** |
>
> As shown in the table above, PoT + RefTool achieves the best average performance. Continued pretraining on the textbook text provides only limited improvement, which suggests that LLMs struggle to directly incorporate textbook knowledge for task solving. Fine-tuning on QA pairs narrows the performance gap, although this approach still requires transforming the textbook content through an LLM, and is more resource-intensive: Fine-tuning must be repeated for every target LLM, while the tools created by RefTool can be reused across models. The experiment is added to the paper.
>
> 3. Human evaluation sample size
>
> > Human evaluation of tool quality uses a small sample—just 20 tools per domain—making it hard to generalize across hundreds created.
>
> We extend the human evaluation size to 50 tools per domain, and the results are shown below.
>
> | **Domain** | **Faithful** | **Function Correct** | **Example Correct** | **Useful** |
> |-----------------|-------------------|---------------------------|--------------------------|-----------------|
> | Causality       |96                | 94                        | 96                      | 92              |
> | Physics         | 90                | 94                        | 100                      | 94              |
> | Chemistry       | 96                | 92                        | 96                       | 90              |
>
> All criteria are satisfied by >=90% tools, showing that most tools are faithfully derived from the references, correctly implemented, and useful in application. The results are updated in the paper.
>
> 4. Cost analysis
>
> > Table 5's cost analysis omits the considerable manual and computational effort required for reference curation and initial tool generation.
>
> Tool generation is fully automated. The LLM creates tools based on the reference content, and verifies and refines the tools automatically. All associated costs are included in the “Toolbox Construction” column in Table 5.
>
> For reference curation, human involvement is minimal. We only download the reference materials from the Internet. For textbooks in PDF format, we convert them to LaTeX automatically, which can be done using free open source tools such as olmOCR [1].
>
> [1] https://github.com/allenai/olmocr

---

> ### Author Response · Authors · 2025-11-20
> **Response to Reviewer m8Qh (2/2)**
>
> 5. Analysis of hierarchical selection
>
> > The hierarchical selection is a critical component. Could you provide metrics on the accuracy of the category and tool selection steps themselves, perhaps against an oracle or human judgment? What is the failure rate at these stages?
>
> Since there is no gold standard category or tool for each question, we compare the degree of agreement between LLMs and human experts. The consistency analysis is in Appendix C, and the results are also shown below.
>
> | **Domain**           | **Consistency**           | **Llama-3.1-70B** | **Gemini-1.5-Pro** | **GPT-4** | **GPT-4o** |
> |----------------------------|--------------------------------|---------------|----------------|-------|--------|
> | Causality | Category Selection             | 100           | 95             | 100   | 100    |
> |                            | Tool Selection within Category | 94            | 100            | 91    | 94     |
> | Physics   | Category Selection             | 80            | 80             | 75    | 76     |
> |                            | Tool Selection within Category | 53            | 56             | 44    | 69     |
> | Chemistry | Category Selection             | 55            | 65             | 72    | 75     |
> |                            | Tool Selection within Category | 60            | 67             | 40    | 90     |
>
> Agreement is higher in category selection than in tool selection, which supports the use of hierarchical selection because it narrows down the search space with high consensus. Across domains, causality shows the strongest consistency in both category and tool selection, owing to more direct questions with keywords like *average treatment effect* that clearly indicates the relevant knowledge. Physics and chemistry questions often involve indirect formulations that make it harder for models to identify the required knowledge.
>
> Gemini-1.5-Pro and GPT-4o demonstrate better alignment with human experts, mirroring their superior PoT performance, which reflects stronger internal domain knowledge. While Llama-3.1-70B and GPT-4 show weaker consistency with humans in physics and chemistry tool selection, their chosen tools are still valuable as relevant knowledge is recalled. In cases where these models select the same category as humans but different tools, we observe a 17% accuracy improvement from tool usage compared to the PoT baseline.

---

> ### Author Response · Authors · 2025-11-27
>
> Dear Reviewer m8Qh,
>
> We would like to kindly ask whether you may have a chance to read our rebuttal and share any follow-up thoughts. Your feedback is very valuable to us, and we are happy to address any further questions.
>
> Best regards, RefTool Authors

---

### Official Review · Reviewer_sMaQ · 2025-10-31

**Soundness:** 3
**Presentation:** 3
**Contribution:** 3
**Rating:** 8
**Confidence:** 4

**Summary:**

REFTOOL introduces a reference-guided framework that enables large language models (LLMs) to automatically create and use tools grounded in external materials (e.g., textbooks, knowledge snippets) rather than relying solely on internal knowledge. This allows LLMs to tackle knowledge-intensive reasoning tasks across domains where pre-defined tools or sufficient prior knowledge are unavailable.
- LLMs generate executable code tools from reference materials (structured like textbooks or unstructured like text snippets). Each tool includes a natural-language description, Python implementation, and example usage. Tools are validated and refined automatically using example-based execution checks.
- Tools are organized into a hierarchical toolbox reflecting the reference’s knowledge structure. During inference, LLMs hierarchically select relevant categories and tools to solve input problems. Integrates with reasoning paradigms like Program-of-Thoughts (PoT) and ReAct, allowing graceful fallback when no tools apply.
- Benchmarked on causality (QRData), physics (TheoremQA), and chemistry (SciBench). Outperforms state-of-the-art tool-creation methods and domain-specific reasoning systems (e.g., Physics Reasoner, ChemAgent) by ~12–13% average accuracy across domains. Achieves dataset-agnostic generalization - tools built from one dataset transfer effectively to others in the same domain. Extends to non-scientific tasks, such as extremely low-resource language translation, yielding a +10 BLEU improvement.
- Reduces computational cost and inference time by up to 99% compared to domain-specific tool systems. Human evaluation shows ≥90% correctness and usefulness of generated tools.

**Strengths:**

- The paper introduces a new perspective on tool creation shifting from internally generated tools (as in Creator, TroVE, etc.) to reference-guided tool generation grounded in external textual materials. This is elegant and underexplored.
- The approach generalizes to multiple datasets within the domain and also to areas beyond scientific reasoning.
- The experiments are extensive, spanning multiple scientific domains and extending to non-scientific domains. The evaluation is robust and covers tool reuse across datasets, ablations, alternative base models for tool creation, cost analysis, and human evaluation.
- The improvements (~12–13%) are consistent and statistically significant.
- The paper is well-written and logically structured, has illustrative figures and examples.
- This work contributes a scalable and general paradigm for extending LLMs’ reasoning ability beyond internal knowledge via reference-grounded code tools. It has clear implications for scientific reasoning, educational AI, and self-improving agents.

**Weaknesses:**

- the validation of generated tools is based on available examples or generation of appropriate examples. It is unclear how one would ensure the correctness of the generated examples and their solution.
- the choice of GPT-4o for tool creation, the number of tools and categories is unclear - what is the basis for these choices?
- While causality, physics, and chemistry benchmarks are informative, they are narrow in scope and format (mostly numerical or formula-based). The framework’s generality would be more convincing if tested on applied or multi-modal reasoning tasks.
- Although the authors claim REFTOOL can extend LLM knowledge boundaries, all experiments use well-bounded domains with relatively small references. It remains unclear whether this approach scales when references are large, diverse, or ambiguous (e.g., thousands of documents).

**Questions:**

covered in the weaknesses above.

---

> ### Author Response · Authors · 2025-11-20
> **Response to Reviewer sMaQ**
>
> Thank you for your helpful comments! Below we address your concerns in detail.
>
> 1. Validation effectiveness
>
> > The validation of generated tools is based on available examples or generation of appropriate examples. It is unclear how one would ensure the correctness of the generated examples and their solution.
>
> To obtain high-quality examples, we encourage the tool creation LLM to use example problems from the reference text whenever such examples exist. The LLM follows this instruction faithfully. We manually examine 20 chemistry tools and find that 50% of them include examples that are taken directly from the textbook, with the corresponding solutions also copied from the original text. Another 25% of the tools contain examples derived from the textbook content, using the same numerical conditions while posing a slightly modified question. This strengthens the reliability of example-based validation.
>
> At the same time, since the tools are created by LLMs, it is difficult to guarantee that every tool is fully correct. Our human evaluation confirms that only a small fraction of tools contain errors. For higher assurance, additional review procedures such as expert examination or multi-round LLM review would be helpful.
>
> 2. Experimental choices
>
> > The choice of GPT-4o for tool creation, the number of tools and categories is unclear - what is the basis for these choices?
>
> We use GPT-4o for tool creation in the main experiments because it was one of the strongest LLMs when we initiated this study. To evaluate robustness with respect to the choice of tool creation model, we also experiment with Llama-3.1-70B-Instruct and Gemini-1.5-Pro, and the results appear in Appendix B.4.
>
> The number of tools and categories is reported in Table 1. For reference materials with a clear structure, such as the scientific textbooks, the categorization follows the original layout of the material. For reference materials without an inherent structure, such as the grammar rules used in the translation experiment, the LLM first generates an appropriate hierarchical structure based on the content. For each segment, we specify only an upper bound on the number of tools that may be created. The actual number is determined by the LLM according to the information contained in that segment.
>
> 3. Generalizability beyond numerical or formula-based reasoning
>
> > While causality, physics, and chemistry benchmarks are informative, they are narrow in scope and format (mostly numerical or formula-based). The framework’s generality would be more convincing if tested on applied or multi-modal reasoning tasks.
>
> Besides the three scientific reasoning domains, we conduct experiments on a natural language reasoning task, **extremely low-resource language translation**, described in **Section 3.3**. This setting is different from the causality, physics, and chemistry benchmarks in scope and format. It is natural language in nature, and the translation outputs cannot be obtained through formulas. Instead, it requires the models to understand and flexibly apply grammatical rules. The tools created from the reference grammar rules help guide the model in selecting and applying the grammar rules. This demonstrates that the RefTool framework extends beyond scientific domains and is effective for non-numerical and non-formula-based reasoning tasks.
>
> 4. Scalability to large or heterogeneous reference corpora
>
> > Although the authors claim REFTOOL can extend LLM knowledge boundaries, all experiments use well-bounded domains with relatively small references. It remains unclear whether this approach scales when references are large, diverse, or ambiguous (e.g., thousands of documents).
>
> Although our experiments focus on well-bounded domains, RefTool can naturally extend to larger and more heterogeneous corpora. A practical approach is to cluster the documents, create tools for each document or cluster, and then combine these tools into a unified toolset. As the quantity of reference materials increases, the toolset can be organized into multiple hierarchical layers. This hierarchical structure allows effective tool selection even when the reference corpus becomes large and diverse.

---

> > ### Comment · Reviewer_sMaQ · 2025-11-28
> >
> > Appreciate the effort from the authors to address my concerns and questions. I maintain my positive outlook on this work with no changes in the scores.

---

> ### Author Response · Authors · 2025-11-27
>
> Dear Reviewer sMaQ,
>
> We would like to kindly ask whether you may have a chance to read our rebuttal and share any follow-up thoughts. Your feedback is very valuable to us, and we are happy to address any further questions.
>
> Best regards,
> RefTool Authors

---

### Official Review · Reviewer_M5bi · 2025-11-03

**Soundness:** 3
**Presentation:** 2
**Contribution:** 2
**Rating:** 2
**Confidence:** 4

**Summary:**

The paper proposes a novel framework that enables large language models (LLMs) to create and use their own tools by leveraging external reference materials (e.g. textbooks, knowledge documents). The approach is motivated by the observation that many complex tasks (causal reasoning, physics, chemistry, etc.) require domain knowledge that may not be encoded in the LLM’s parameters. Unlike prior tool-using paradigms, which rely purely on the model’s internal knowledge or a fixed set of APIs, REFTOOL “grounds” the tool creation process in external references.

**Strengths:**

1.	Enabling LLMs to automatically create executable tools from unstructured text references is more efficient than traditional text retrieval. This structured generation approach (including pseudocode and code templates) is more effective than simple text retrieval. By transforming textbooks into a toolbox, REFTOOL empowers models to solve problems previously beyond their capabilities.
2.	REFTOOL achieves significant performance improvements on knowledge-intensive benchmarks like causal reasoning, physics, and chemistry, increasing accuracy by 12.3% compared to existing tool-generation methods.

**Weaknesses:**

1.	The method currently relies on GPT-4 (or similar large models) for tool generation and verification. While understandable (tool synthesis is difficult), this means the initial setup can be expensive and dependent on proprietary models. Without GPT-4 or similar high-performance models, the quality of the generated tools may decline. This introduces an unfairness in experimental baseline comparisons. The article does not deeply explore using open-source LLMs for tool creation; it would be worth knowing how a 70B open-source model performs on this task, even if some performance loss is expected.
2.	Although overall tool quality is high, the generated tools might be relatively unreliable in some highly complex domains (like chemistry). For instance, chemical reasoning may require extremely complex domain knowledge, and even with textbooks and GPT-4, generating completely correct code isn't guaranteed. The article notes that chemistry tools received relatively low scores in human evaluation. Therefore, for highly specialized or obscure knowledge, the method might require additional fine-tuning (perhaps incorporating symbolic systems or domain expert review).
3.	REFTOOL generates hundreds of tools per textbook (e.g., over 500 for physics). Storing and managing such a large toolbox could become overwhelming, especially when considering broader knowledge bases or multiple textbooks. Although hierarchical selection helps, it can become quite complex – for example, if a category still contains dozens of tools, providing all their descriptions to the LLM might exceed its context window limit.

**Questions:**

1.	The article's innovation seems extremely limited relative to existing work [1], essentially just pre-generating tools in a RAG-like manner. What do the authors believe is the innovative contribution of their research?
[1] Cai, T., Wang, X., Ma, T., Chen, X., & Zhou, D. Large Language Models as Tool Makers. In The Twelfth International Conference on Learning Representations.

---

> ### Author Response · Authors · 2025-11-20
> **Response to Reviewer M5bi (1/3)**
>
> Thank you for your constructive comments! Below we address your concerns in detail.
>
> 1. Using open-source LLMs for tool creation
>
> > The article does not deeply explore using open-source LLMs for tool creation; it would be worth knowing how a 70B open-source model performs on this task, even if some performance loss is expected.
>
> To validate the robustness of the tool creation module, we experiment with Llama-3.1-70B-Instruct and Gemini-1.5-Pro as the tool creation LLMs in Appendix B.4. The results are also shown below.
>
> Physics:
> | **Method** | **Llama-3.1-70B** | **Gemini-1.5-Pro** | **GPT-4**     | **GPT-4o**    | **Average**   |
> |--------------------------------------|-------------------|--------------------|---------------|---------------|---------------|
> | Physics Reasoner                     | 48.2              | 50.9               | 42.1          | 33.3          | 43.4          |
> | LATM                            | 38.9          | 33.3          | 39.8          | 30.6          | 35.7          |
> | Creator                         | 40.4          | 57.0          | 35.1          | 40.4          | 43.2          |
> | TroVE                           | 33.3          | 58.8          | 35.1          | 48.2          | 43.9          |
> | PoT                                  | 48.2              | 57.9               | 45.6          | 57.0          | 52.2          |
> | PoT + RAG                            | 44.7              | 57.0               | 44.7          | 57.9          | 51.1          |
> | PoT + RefTool (GPT-4o)      | **53.5**     | 58.8               | 49.1          | 57.9          | **54.8** |
> | PoT + RefTool (Gemini)      | 48.2              | 58.8               | 47.4          | **58.8** | 53.3          |
> | PoT + RefTool (Llama)       | 49.1              | **60.5**      | **50.0** | 57.0          | 54.2          |
>
> Chemistry:
> | **Method** | **Llama-3.1-70B** | **Gemini-1.5-Pro** | **GPT-4**     | **GPT-4o**    | **Average**   |
> |--------------------------------------|-------------------|--------------------|---------------|---------------|---------------|
> | LATM                            | 31.4          | 25.1          | 45.0          | 35.7          | 34.3          |
> | Creator                         | 40.1          | 60.0          | 46.9          | 43.3          | 47.6          |
> | TroVE                           | 38.6          | 65.6          | 39.2          | 52.7          | 49.0          |
> | StructChem                           | 37.9              | 50.2               | 29.7          | 40.5          | 39.6          |
> | ChemAgent                            | 48.2              | 65.5               | 52.5          | 58.9          | 56.3          |
> | PoT                                  | 46.9              | 62.3               | 51.8          | 58.9          | 55.0          |
> | PoT + RAG                            | 48.1              | 63.7               | **53.7** | 56.6          | 55.5          |
> | PoT + RefTool (GPT-4o)      | **49.5**     | **66.4**      | 53.4          | 61.3          | **57.7** |
> | PoT + RefTool (Gemini)      | 48.3              | 64.9               | 52.5          | **61.6** | 56.8          |
> | PoT + RefTool (Llama)       | **49.5**     | 62.4               | 53.2          | 57.3          | 55.6          |
>
> Across both domains, using these LLMs for tool creation still yields performance superior to all baseline methods, demonstrating that RefTool is robust to the choice of the base model and fully compatible with open-source LLMs, which is particularly important when working with sensitive or proprietary reference materials.

---

> ### Author Response · Authors · 2025-11-20
> **Response to Reviewer M5bi (2/3)**
>
> 2. Tool quality in complex domains
>
> > Although overall tool quality is high, the generated tools might be relatively unreliable in some highly complex domains (like chemistry).
>
> To further investigate the reliability of tools, we extend the human evaluation to 50 tools for each domain, and the results are shown below.
>
> | **Domain** | **Faithful** | **Function Correct** | **Example Correct** | **Useful** |
> |-----------------|-------------------|---------------------------|--------------------------|-----------------|
> | Causality       |96                | 94                        | 96                      | 92              |
> | Physics         | 90                | 94                        | 100                      | 94              |
> | Chemistry       | 96                | 92                        | 96                       | 90              |
>
> Although chemistry scores are slightly lower than the other domains, the tools still satisfy all quality criteria in ≥90% of cases. This indicates that even in highly technical scientific domains, the generated tools are generally faithful, correctly implemented, and practically useful.
>
> Below is an example tool created from the final chapter of the chemistry textbook, implementing the Butler–Volmer equation for electrode current density:
> ```
> def butler_volmer_current_density(j0, alpha, eta, T=298.15):
>     """
>     Calculate the current density at an electrode using the Butler-Volmer equation.
>     Parameters:
>     - j0 (float): Exchange-current density (A/cm^2).
>     - alpha (float): Transfer coefficient (dimensionless, typically between 0 and 1).
>     - eta (float): Overpotential (V).
>     - T (float): Temperature in Kelvin (default is 298.15 K).
>     Returns:
>     - float: Net current density (A/cm^2).
>     """
>     import math
>     from scipy.constants import R, e
>     # Calculate f = F / (RT), where F (Faraday constant) = e * Avogadro's number
>     F = e * 6.02214076e23  # Faraday constant in C/mol
>     f = F / (R * T)
>     # Butler-Volmer equation
>     j = j0 * (math.exp((1 - alpha) * f * eta) - math.exp(-alpha * f * eta))
>     return j
> ```
> Even though this involves nontrivial electrochemistry knowledge, GPT-4o produces a correct tool guided by the textbook.
> While current LLMs understand the reference materials we provide quite well, there will be cases where the content demands deeper, domain-specific expertise to fully understand. In such cases, additional fine-tuning or incorporating expert knowledge can further improve tool quality. Also, with the increase of LLM capability, it is very likely for the LLMs to understand more advanced reference materials about expert knowledge and transform the key knowledge into useful tools.
>
> 3. Toolset size and management
>
> > REFTOOL generates hundreds of tools per textbook (e.g., over 500 for physics). Storing and managing such a large toolbox could become overwhelming, especially when considering broader knowledge bases or multiple textbooks. Although hierarchical selection helps, it can become quite complex – for example, if a category still contains dozens of tools, providing all their descriptions to the LLM might exceed its context window limit.
>
> In our toolsets, each category contains at most 18 tools, amounting to at most 5,427 tokens (measured by the GPT-4o tokenizer). This is well within the context windows of modern LLMs (≥128k tokens).
>
> Moreover, it is common for toolsets to contain a large number of tools. For example, ToolBench [1] has 16,464 tools, and UltraTool [2] has 2,032 tools. Compared with them, our toolset is relatively small, and modern LLMs are capable of handling toolsets of this scale.
>
> If RefTool is extended to multiple textbooks or broader knowledge bases, the hierarchy can naturally add higher layers (e.g., discipline → textbook → chapter), preserving scalability.
>
> [1] Qin, Yujia, et al. "ToolLLM: Facilitating Large Language Models to Master 16000+ Real-world APIs." The Twelfth International Conference on Learning Representations.
>
> [2] Huang, Shijue, et al. "Planning, Creation, Usage: Benchmarking LLMs for Comprehensive Tool Utilization in Real-World Complex Scenarios." ACL (Findings). 2024.

---

> ### Author Response · Authors · 2025-11-20
> **Response to Reviewer M5bi (3/3)**
>
> 4. Innovative contributions relative to LATM
>
> > The article's innovation seems extremely limited relative to existing work (LATM), essentially just pre-generating tools in a RAG-like manner. What do the authors believe is the innovative contribution of their research?
>
> While both LATM and our method involve LLM-based tool creation, their methodologies and capabilities differ substantially.
>
> LATM is designed for tasks with fixed, rule-based structures (e.g., word sorting, logical deduction). Its tools are crafted from a few demonstrations and rely heavily on the model’s internal knowledge. As our experiments show, LATM performs poorly on scientific reasoning.
>
> In contrast, we propose a general way to create tools based on reference materials, with the innovative contributions:
>
> - **Grounding tool creation in external reference materials.** Previous approaches depend on the LLM’s built-in knowledge and cannot handle information outside its training data. REFTOOL instead uses external references to create tools, enabling LLMs to reason beyond their inherent knowledge boundaries.
>
> - **Hierarchical organization of tools.** Previous tool creation methods do not address tool organization, which may cause selection difficulty as the toolset grows. RefTool introduces hierarchical structuring following the reference material (or generated by the LLM for unstructured materials), making tool selection efficient and scalable.
>
> - **Effectiveness across domains.** RefTool achieves significantly stronger performance on scientific reasoning than previous tool creation methods, with 13% accuracy gain on average. It also generalizes to natural-language tasks such as extremely low-resource translation.
>
> More broadly, we believe it is both promising and necessary to ground LLM reasoning in external materials. Even as LLMs grow stronger, they will always encounter knowledge they do not inherently possess, for example, newly released Python packages or methods introduced in recent scientific papers. Our experiments show that simple RAG methods are not enough to bridge this gap, whereas tools provide an effective medium for integrating external knowledge into LLM reasoning.

---

> ### Author Response · Authors · 2025-11-27
>
> Dear Reviewer M5bi,
>
> We would like to kindly ask whether you may have a chance to read our rebuttal and share any follow-up thoughts. Your feedback is very valuable to us, and we are happy to address any further questions.
>
> Best regards,
> RefTool Authors

---

### Author Response · Authors · 2025-12-03
**Rebuttal Summary**

We thank all reviewers for their thoughtful and constructive reviews!

Strengths:
- Reviewers `sMaQ` and `PQMZ` highlight the *conceptual novelty* of RefTool: shifting from internally generated tools to reference-guided tool creation grounded in external materials. Reviewer `sMaQ` further emphasizes that it contributes a scalable and general paradigm for extending LLMs’ reasoning ability, with clear implications for scientific reasoning, educational AI, and self-improving agents.
- All reviewers praise the *methodological design*. They find the framework methodologically sound with strong robustness mechanisms (Reviewer `PQMZ`) and elegant in its formulation (Reviewer `sMaQ`). Reviewer `m8Qh` echoes that grounding tools in authoritative sources is a principled and well-motivated design decision, and Reviewer `M5bi` highlights that the hierarchical toolbox structure that mirrors the organization of the reference material is a strong design choice.
- Reviewers `M5bi`, `sMaQ`, and `PQMZ` commend the *empirical performance* of RefTool, noting that it consistently achieves substantial gains over both strong tool-creation baselines and domain-specialized methods. Reviewers `sMaQ`, `m8Qh`, and `PQMZ` also appreciate the *generalizability* of RefTool, especially its effectiveness on a non-scientific task using unstructured reference materials, demonstrating the method’s flexibility and broad applicability.

We carefully address all reviewer comments in the rebuttal, and summarize key additions and clarifications below:
- We clarified that experiments using open-source LLMs for tool creation, as well as the human evaluation of hierarchical selection, are included in the Appendix.
- We added baselines of finetuning LLMs on reference materials, showing that these baselines fall short of RefTool while being more resource-intensive.
- We expanded the human evaluation to 50 tools per domain and evaluated Gemini-generated tools, demonstrating consistently high tool quality across models.
- We added an experiment generating prerequisite tools to support smaller LLMs, showing that when models lack foundational domain knowledge, such prerequisite tools serve as an effective complement to enable more reliable tool use.

---

### Meta-Review · Area_Chair_UCj9 · 2026-01-01

**Summary:**

The paper introduces a new for tool generation using structured reference materials into a hierarchical tool organization framework that LLMs can apply during inference for answering questions that require domain specific reasoning. The authors first describe their method and then evaluate it against a set of baselines consisting mostly of previous tool generation methods and ablation of their method. The paper also contains an analysis of generalizability with a translation task along with an analysis probing tool re-usability, inference cost and tool quality as judged by human annotators.

The reviewers generally praised the novelty of the method as a new work to perform tool generation based on high-quality reference materials in challenging domains, such as scientific domains. The reviewers also praised the experiments and analysis performed by the authors, which ultimately show the advantages of the method compared to related work and baselines. Some reviewers also praised the organization of the paper and the demonstration of generalizability of the framework.

The main weaknesses called out by the reviewers were related to providing additional clarification on the generated tools and the scalability and importance of the hierarchical tool organization method. These questions were mostly addressed in the rebuttal, as acknowledged by some reviewers. The authors also performed additional experiments with a fine-tuning baseline to further understand the utility of the RefTool framework.

While the authors addressed many of the reviewer's queries, remaining limitation relates to the use of GPT-4o as the main tool generation model that adopts the hierarchical framework and the assessment of tool quality at scale. In the reviewer discussion, the authors acknowledge the limitation that a powerful LLM is initially required and that tool assessment at scale remains a challenge. My recommendation is that the author describe these limitations explicitly in a revision and potentially provide some ideas for future work along those dimensions.

The assessment of generalizability is what some reviewers called out as a potential weakness asking for additional analysis and experiments to that end.

**Reviewer Concerns:**

Addressed Concerns:
* Reviewer M5bi's concern on prior work and tool management were addressed. Reviewer M5bi's concerns on using GPT-4o as a base model and providing tool quality assessment were mostly addressed with some limitations acknowledge.
* Reviewer sMaQ's concerns were mostly addressed.
* Reviewer m8Qh's concerns were mostly addressed.
* Reviewer PQMZ's concerns were mostly addressed with one important limitation on using a powerful base model acknowledged by the authors.

Outstanding concerns:
* Reviewer M5bi's concerns on using a powerful base model and doing tool quality assessment at scale. This limitation was acknowledged by the authors, including in discussions with Reviewer PQMZ.
* Reviewer sMaQ's concerns were mostly addressed.
* Reviewer m8Qh's concerns were mostly addressed.
* Reviewer PQMZ's concern on the dependence on GPT-4o as critical to performance was acknowledged as a limitation.

**Reviewer Scores:**

* Reviewer M5bi's would raise score to 4.
* Reviewer sMaQ maintains score at 8.
* Reviewer m8Qh raises score to 6.
* Reviewer PQMZ maintains score at 6.

---

### Decision · Program_Chairs · 2026-01-26

Accept (Poster)